# Optimisation of Mix Proportion of 3D Printable Mortar Based on Rheological Properties and Material Strength Using Factorial Design of Experiment

**DOI:** 10.3390/ma16041748

**Published:** 2023-02-20

**Authors:** Sandipan Kaushik, Mohammed Sonebi, Giuseppina Amato, Utpal Kumar Das, Arnaud Perrot

**Affiliations:** 1School of Natural and Built Environment, Queen’s University Belfast, Belfast BT7 1NN, UK; 2Department of Civil Engineering, Tezpur University, Napaam, Tezpur 784028, India; 3FRE CNRS 3744, IRDL, Université Bretagne Sud, 56100 Lorient, France

**Keywords:** 3D printable mortar, basalt fibre, rheology, compressive and flexural strength, factorial design, isoresponse curve

## Abstract

In the production of 3D printable mortar (3DPM), numerous efforts have been made globally to effectively utilise various cementitious materials, admixtures, and fibres. The determination of rheological and material strength properties is crucial for successful 3D concrete printing because the materials used in 3DPM must possess the unique characteristic of making mortar flowable while being strong enough to support the weight of subsequent layers in both fresh and hardened states. The complexity of the required characteristics makes it challenging to develop an optimised mix composition that satisfies both the rheological and material strength requirements, given the wide range of available admixtures, supplementary cementitious materials, and fibres. Fly ash, basalt fibre and superplasticiser when blended with cement can help to improve the overall performance of 3DPM. The objective of this research is to optimise the rheological properties and material strength of 3D printable mortars (3DPM) containing cement, fly ash, basalt fibre, and superplasticiser. This study aims to produce 3DPM with an optimised mix composition to meet the requirements of both rheological and material strength characteristics using the factorial design approach and desirability function. Different dosages of cement, fly ash, basalt fibre, and superplasticiser are chosen as the primary design parameters to develop statistical models for the responses of rheological and material strength properties at 7 and 28 days. The results expressed in terms of the measured properties are valid for mortars made with cement ranging from 550 to 650 kg/m^3^, fly ash from 5% to 20% (of cement), superplasticiser from 2 to 4 kg/m^3^, and basalt fibre from 1 to 3 kg/m^3^. The rheological properties are evaluated using slump flow, cone penetrometer, and cylindrical slump tests, while the mechanical strength is evaluated using a three-point bending test and compressive test. A full factorial design experiment (FoE) is used to determine the significant parameters effecting the measured properties. Prediction models are developed to express the measured properties in terms of the primary parameters. The influence of cement, fly ash, basalt fibre, and superplasticiser is analysed using polynomial regression to determine the main effects and interactions of these primary parameters on the measured properties. The results show that the regression models established by the factorial design approach are effective and can accurately predict the performance of 3DPM. Cement, fly ash, and superplasticiser dosages have significant effects on the rheological and mechanical properties of mortar, while basalt fibre is able to influence the static yield stress and flexural strength of 3DPM. The utilisation of regression models and isoresponse curves allows for the identification of significant trends and provides valuable insight into the behaviour of the material, while desirability function is useful to optimise overall performance of mix proportions to meet the desired performance objective at fresh and hardened states.

## 1. Introduction

Three-dimensional concrete printing (3DCP) is one of the most transformative developments for the construction industry in recent years. By utilising 3DCP technology, the need for traditional formwork is eliminated, leading to significant time, cost, and labour savings [1,2]. A majority of the 3DCP procedures involve the use of a nozzle, often mounted on a gantry or robotic arm, to carefully deposit cementitious material throughout the construction process [2]. It is important to note that, without the use of formwork, the structure is at significant risk of deforming during the initial curing stage due to the absence of a support system [3]. Therefore, both extrudability, i.e., the ability of the cementitious material to be extruded via a nozzle with ease, and buildability, i.e., the ability to remain rigid enough to sustain the weight of subsequent layers, are critical to the success of the 3D concrete printing process [2,4,5,6]. Developing a cementitious mixture that meets the narrow range of rheological requirements for both extrudability and buildability is vital for reliable 3D printing and the wider adoption of this technology in the construction industry.

The 3DCP entails a sequence of processes (pumping, extrusion, and deposition of subsequent layers) with distinct physical principles, imposing stricter rheological constraints than typical concrete construction methods [3,4,7,8]. Although the associated rheological properties are the same across these processes, their desired values change. To ensure that a cementitious mix is flowable and consistent throughout the printing process while having appropriate strength to sustain successive layers at an early age [9,10], it is vital to examine the rheological properties prior to material deposition. Rheological requirements can vary between 3D printers due to differences in printing processes and material properties [6,7]. To ensure consistent and successful printing, it is necessary to tailor the mix design of the chosen materials to the specific operational demands of the printer for maintaining good extrudability and buildability, which are governed by properties such as slump, slump flow, penetration resistance, and yield stress of fresh concrete [3,9,10]. The ease of extrusion and buildability in 3D concrete printing is often influenced by the flowability and static yield stress of the cementitious material [4,11,12,13,14]. Therefore, it is important to consider both of these factors in order to accurately characterise the extrudability and buildability of the 3DCP process. Existing research has reported measuring plastic viscosity and yield stress using a rheometer for various mixtures of 3DCP [2,11,15]; however, much like the printing process, these rheological tests conducted on the cementitious mixtures are highly dependent on the measuring methodology and equipment used [16,17]. In this study, field-friendly tests such as slump flow, cylindrical slump test, and cone penetration tests are used to measure the rheology of mortar mix. While several studies have explored the influence of mix proportions on extrudable and buildable concrete [11,18,19,20,21,22], few have systematically sought to optimise the mix. Despite several empirical mix design studies for 3DCP [9,12,23,24], there are no universally recognised mix design approaches, and few of these research studies incorporated material strength at hardened state for optimisation of mix design. A factorial design of experiments (FoE) and an ANOVA (analysis of variance) are used to create predictive models that fit the experimental results and streamline the experimental process, reducing the number of trials. A desirability function is also introduced to achieve the desired values of measurable properties through the simultaneous optimisation of multiple objectives.

Statistical models can be used to predict how different materials influence the rheological behaviour and strength of 3D printable concrete, which is now made with a wider range of materials in addition to cement. This approach is more efficient than the traditional experimental design, in which only one variable is changed at a time. The use of 3DCP in construction has been criticised for its high use of Portland cement, which is known to have a significant environmental impact [6]. However, 3DCP can be made more environmentally friendly by adding supplementary cementitious materials (SCMs) such as fly ash, GGBS, and silica fume to partially replace cement in the mix [12,25,26,27]. In addition, high range water reducers (HRWR), superplasticisers, early strength additives, accelerators, and other such admixtures are also used to achieve the desired rheology or improve the working performance of 3DCP mixtures. Studies have shown that incorporating large amounts of fly ash can make 3DCP more easily extrudable without significantly increasing its static yield stress [26,28]. Dispersing and water reducing agents known as superplasticisers, such as polynaphthalene sulfonate (PNS) and polycarboxylate ether (PCE), are often used in 3DCP mixtures to improve flowability and reduce the amount of water needed [29,30,31]. Most 3DCP mixtures do not contain coarse aggregates, which can increase the packing fraction of the material and improve its apparent yield stress, or the amount of force required to cause it to flow [32]. However, superplasticisers work by reducing the surface tension of the water in the mixture and allowing the particles to flow more easily, which can decrease the apparent yield stress and partially counter the effect of the increased packing fraction [28,29]. PCE is particularly effective at improving flowability because it reduces plastic viscosity through steric hindrance, which reduces attraction forces between particles and prevents them from coming into close contact [28,29,31]. Unlike superplasticisers, fibre is known to increase the yield stress of concrete [33,34]. The addition of fibres to concrete can significantly increase its flexural and tensile strength, but it is important to carefully control the amount of fibre used [35,36,37]. According to research conducted by Le et al. [12], a fibre content of 1.2 kg/m^3^ is sufficient for producing an optimised mix of printable concrete using micro polypropylene fibres. While higher amounts of fibre can improve the buildability due to the high yield stress, they can also hinder extrudability. Using locally sourced materials from the UK, such as basalt rock to produce basalt fibre [38], researchers improved flexural strength in basalt fibre-reinforced printed elements by fibre-reinforcing the cementitious matrix or increased the flexural toughness of high-performance fibre-reinforced concrete (HPFRC) [36,39].

The success of 3D concrete printing relies heavily on the selection of raw materials and the design of the mixture to ensure appropriate extrudability and buildability of the material [12,40]. These characteristics are crucial in determining ability of the material to flow through the printing nozzle, maintain its shape after extrusion, and have the necessary yield stress to the support subsequent layers during the printing process. Mix design for 3D concrete printing has traditionally been limited to selecting a few raw materials and optimizing their proportions through trial-and-error methods. However, the integration of factorial design into the mix design process provides a more comprehensive and efficient approach to selecting and blending raw materials by simultaneously considering multiple factors and their interactions. As such the objective of the research presented in this paper is to provide an approach which can optimise multiple goals to improve extrudability, buildability and material strength of hardened mortar utilizing a suitable mix proportion. The concept of desirability function is used in this paper for multi-objective optimisation of measured properties. At present, there is no standard method to quantify the extrudability and buildability of 3D printable mortar. Previous research has explored the use of multiple tests to characterise the rheological properties of 3D concrete printing mixtures [6,9], but few studies have utilised field-friendly tests such as the flow table, cone penetration, or slump test to assess extrudability and buildability. This study aims to fill this gap by optimising the mix proportion of printable mortar using the results from the flow value, penetration depth, and static yield stress derived from the slump test to evaluate extrudability and buildability. Establishing a clear relationship between the rheological parameters of fresh concrete and its composition is crucial to guide the mix design of printable mortar. While some models have been developed for this purpose [9,22], further refinement is needed to make them reliable tools. Currently, mix design for 3DPM primarily focuses on ensuring the printability of fresh concrete, but little attention has been paid to the characteristics of hardened concrete, such as compressive and flexural strength. Further research is required to develop mix design tools that target the desired properties of both fresh and hardened concrete. This paper attempts to provide a solution for developing a wholesome optimisation approach to incorporate essential fresh and hardened properties of 3DPM.

The aim of this study is to optimise the mix proportion of four key materials, cement, superplasticiser, basalt fibre, and fly ash, to maximise the extrudability, buildability, and material strength of printed mortars. The optimisation process involves using statistical modelling (FoE) to evaluate the effects of varying dosages of these materials on the fresh and rheological properties, as well as the hardening behaviour of cement mortars. The properties of the proposed mixes are assessed through a range of methods, including flow table, cone penetration, cylindrical slump, compressive test, and three-point bending tests on printed and mould-cast specimens. The results of the experimental work are analysed and used to develop statistical models that accurately estimate the rheological and strength characteristics of the materials. These models identify the primary parameters and two-way interactions that have a significant impact on the fresh and hardening properties of mortars, making it easier to assess the potential impact of varying the dosages. By streamlining the mix design process, the proposed models can simplify the optimisation process and reduce the number of trial batches required compared with the conventional one-factor-at-a-time method. To maintain both the number of measured properties and the scope of the optimisation process, both compressive and flexural strengths are measured perpendicular to the direction of printing only.

## 2. Materials and Methods

### 2.1. Materials

The cement used in this study is Portland cement (CEM) type CEMI 52.5 N, as specified by BS EN 197-1: 2011 [41], and has a specific gravity of 3.10. The fly ash with a specific gravity of 2.21 was supplied by Scot Ash Ltd. and conforms to BS EN 450-1:2012 [42]. The chemical composition and physical properties of cement and fly ash can be found in Table 1 and Table 2, respectively. As the water-binder (W/B) ratio is known to have significant long-term influence on the strength of cementitious material, it is kept constant at 0.41 for all the mixtures in order to study the influence of cement and fly ash as binder on material strength independently. A polycarboxylate-based superplasticiser with a specific gravity of 1.06 is used to increase workability in fresh state. Basalt rock is a single-component natural resource that is ideal for the production of continuous basalt fibre due to its low environmental impact. Continuous basalt fibre is a product made in a single step from molten basalt rock without the use of any chemicals. Chopping continuous basalt fibre into short lengths produces basalt fibre [43]. Basalt fibre (BA) used in the mortar mixture was provided by Basalt Technologies UK Ltd. and it has a specific gravity of 2.65 and a length of 12.7 mm. Sand with a maximum particle size of 1.18 mm, absorption coefficient of 0.8%, and specific gravity of 2.7 was used. Quantity of sand was adjusted to maintain a constant volume. The grain size distribution of the sand can be seen in Figure 1.

### 2.2. Factorial Design of Experiment

The statistical approach known as “factorial design of experimental” is used to investigate the interaction of several parameters simultaneously. Dosages of input parameters (selected materials) in the mortar mixtures were varied to assess their potential influence on the measured responses. This technique not only identifies the major parameters that influence the measured response but also the mechanism through which these parameters do so [44] (pp. 238–248). Polynomial regression model for a two-level factorial design is given by Equation (1) and the regression model considers the interactions between two parameters, in addition to the main effects.
(1)$$y=\beta_{0}+\overset{j}{\underset{i=1}{\sum }}\beta_{i}x_{i}+\underset{i}{\sum }\underset{j}{\sum }\beta_{ij}x_{i}x_{j}\;\;\;\;where\;\;i<j$$

In Equation (1), *x_i_* and *x_j_* represent the material concentration in coded terms, *β*_0_ denotes the offset term, and *β_i_* and *β_ij_* represent the regression coefficients for main effect and interaction effect, respectively. The relationship between material concentration in coded terms (*x_i_*, *x_j_*) and actual terms (*Conc*) is:(2)$$x_{i},x_{j}=\frac{Conc-\left\{\left(Conc_{low}+Conc_{high}\right)/2\right\}}{\left\{\left(Conc_{high}-Conc_{low}\right)/2\right\}}$$

In this study, the factorial design is used to quantify the rheological and mechanical properties of fibre reinforced cementitious materials used in 3D concrete printing. The cementitious material in this experiment is made up of four key raw materials: cement (CEM), superplasticiser (SP), basalt fibre (BA), and fly ash (FA), which are deemed to have a significant influence on the properties of cementitious mixture and are chosen to formulate statistical models for evaluating rheological properties and material strength. The 2^k^ (k = 4) full factorial design approach, where k represents the number of parameters or raw materials, is used to evaluate the influence of these materials used in the cementitious mixtures. The analysis of the experimental findings will lead to the identification and quantification of the parameters and two-way interactions that significantly affect the properties of fresh and hardened cement mortars. The derived statistical equations will be used to evaluate the impact of changing the dosage contents. The dosages of cement (CEM), fly ash (FA), superplasticiser (SP), and basalt fibre (BA) can be changed to ensure the stability of the mortar proportions. The proposed models can make it easier to test how to improve a given mix by reducing the number of trial batches needed to find the best balance between the different mix variables. The derived statistical models are valid for mixes made with cement (CEM) ranging from 550 to 650 kg/m^3^, percentage of fly ash (FA) from 5 to 20% by mass of cement, the dosages of basalt fibre (BA) from 1 to 3 kg/m^3^, and dosages of superplasticiser (SP) from 2 to 4 kg/m^3^. As reported in a review of multiple extrusion-based concrete printing mixtures [6], the binder content used in cementitious mixtures is typically 600 kg/m^3^ or more (in some cases reaching up to 1100 kg/m^3^). To ensure required performance of printable concrete, including rheology, coherent adhesion and early strength, a higher content of Portland cement has been adopted as a general solution [45]. This high binder content, which is commonly the most energy-intensive component of concrete, renders 3D concrete printing less environmentally friendly due to its substantial energy consumption. The inclusion of a low fly ash (FA) replacement percentage of 5% in the experiment was aimed at comprehending the lower limits of fly ash usage and its impact on the mixture’s properties. The upper limit of fly ash content in the mixture was limited to 20% due to concerns of its impact on the compressive strength development of the material over a 28-day period. Table 3 presents a summary of the different values used for each of the parameters.

A total of twenty selected mixes considered in the factorial design are listed in Table 4. The first sixteen mixtures consist of all possible combinations of the minimum and maximum values of the four factors considered. Additionally, a mix at the central point was replicated four times to estimate the experimental error and examine the reliability of the models. Mixtures from 17 to 20 are centre points with the same properties that are the average values of the four factors. The cementitious material was tested with flow table, cone penetration, cylindrical slump, compression test, and three-point bending test for flexural strength.

### 2.3. Testing Procedure

#### 2.3.1. Mixture Preparation

A constant water-to-binder ratio (w/b) of 0.41 was used for all the mortar mixtures. Water temperature was maintained at 20 ± 1 °C and mixtures were prepared in 6 min in a planar-action high shear mixer with maximum capacity of 5 L. Portland cement (CEM), fly ash (FA) and sand were mixed for 1 min at low speed (140 rpm) before adding a solution of water and SP. The time when water and SP was added is considered as zero time. The mortar was further mixed for 1 min at a low speed. After mixing for 1 min, the mixer was stopped, and any lumps of solids were broken and dry mixture along the circumference of the mixer was thoroughly mixed before adding basalt fibre (BA). The mortar was then mixed for 2 min on high speed (285 rpm) and 1 min on low speed (140 rpm).

#### 2.3.2. Flow Table Test

The timing and sequencing of all test procedures were the same for all mixes. The flow table test conforming to BS EN 1015-3:1999 [46] was used to assess flowability (6 min after starting mixing). The fresh mortar was conventionally cast in a truncated cone shaped mould with a bottom diameter of 100 mm, an upper diameter of 70 mm, and height of 60 mm, placed at the centre of a jolting table. The cone was filled with mortar in two layers and each layer was compacted by at least 10 short strokes of the tamping rod to ensure uniform filling of the mould. After removing the conical mould, the table was jolted 15 times at regular intervals. At the end of jolting, the flow was stopped and the spread of the mortar was measured in two perpendicular directions to obtain the average flow value. The flow value obtained from the flow table test is a measure of the fluidity of the fresh mortar and is directly associated with the extrudability of the mortar when it is printed through the nozzle attached to the ram extruder. A high flow value obtained from the flow table test is indicative of the mortar’s ability to flow with ease, which implies that it has improved extrudability.

#### 2.3.3. Cone Penetration Test

The purpose of this test is to measure the workability and consistency of the mortar under a certain stress caused by the descending cone (under gravity). At first the conical mould with a bottom diameter of 80 mm, an upper diameter of 70 mm, and height of 40 mm was conventionally cast with fresh mortar in two layers. Then each layer was compacted by at least 10 short strokes to ensure uniform filling of the mould. The conical plunger of the penetrometer conforming to BS ISO 13765-1:2004 [47] was adjusted so that it touched the surface of the mortar before releasing, and the digital reading in the screen of the penetrometer was noted down as the initial reading. The conical plunger was then allowed to sink inside the mortar for 5 s under its own weight and the final reading was noted down. The difference of the reading gives the actual penetration value of the mortar. An average of three readings was reported to better represent the consistency of each mixture. The penetration value of the mixture is an indicator of its fluidity or workability and can be directly linked to the extrudability of the material. As the penetration depth increases, the fluidity and workability of the mixture increase, leading to improved extrudability. A low penetration value suggests that the mortar mixture is less workable, making it harder to extrude, while a high penetration value implies that the mixture is more fluid and can be easily extruded.

#### 2.3.4. Cylindrical Slump Test

The slump test is an empirical test [48] performed on mortar to quantify the yield stress and to assess the spread and slump of the mortar. Because the mix needs to be stiff for printing, a typical rheometer would not be able to measure the yield stress of the investigated mortars. A cylindrical slump mould with an interior diameter of 64 mm, an exterior diameter of 66 mm, and a height of 136 mm was placed at the centre of a plane table. Mortar was then filled into the mould in two layers; each layer was compacted with 10 short strokes with the help of a tamping rod to ensure a uniform distribution of material and reduce air pockets in the mould. The excess mortar was scraped off with a trowel and the mould was lifted after approximately 15 s. Then two height and diameter values in two perpendicular directions were measured.

These values were used in estimating the static yield stress (*τ_y_*) of the mortar depending on the ratio height/spread. There are three main flow regimes with considerably different stress tensors depending on the observed values and these flow regimes were represented as H >> R, H ≈ R and H << R where H denotes the height and R denotes the radius after lifting the mould [48,49]. A static yield stress calculation formula was established for each regime. In this experiment, most mortars were found to follow the H >> R regime, while a few followed the H ≈ R regime. To calculate the yield stress in both regimes, two dimensionless quantities stress (*τ_y_*′) and strain (*s*′) were used.
(3)$${\tau_{y}}^{\prime }=\frac{\tau_{y}}{\rho_{m\;}\times g\times h_{0}}\;$$
(4)$$s^{\prime }=\frac{s}{h}\;\;$$
where stress (*τ_y_*′) represents ratio of yield stress (yielding under the force of gravity) after the mould is removed to the yield stress when the material is confined in the mould. *τ_y_* is the yield stress, *ρ_m_* is density, g is acceleration due to gravity, h_0_ is the initial height, *s* is the slump, h is the height after the cylindrical mould is lifted, and strain (*s*′) is the ratio of slump (*s*) to height (*h*).

In this experiment the dominant regimes can be represented as “Pure elongational flow (H ≫ R)” and “Intermediate (H ≈ R)” [48] and yield stress for these regimes can be obtained by using the following expressions (3) and (4).

Case 1: Pure elongation flow regime

In this case the radial stress variations are negligible in comparison to the vertical stress variations and dimensionless stress quantity is obtained using the equation.
(5)$${\tau_{y}}^{\prime }=\frac{1-s^{\prime }}{\sqrt{3}}\;$$

Case 2: Intermediate regime

Considering the case of ideal elastic solid when maximum stress is applied in normal direction [50] (gravity), the simple expression for yield stress can be written as a function of slump using the equation.
(6)$${\tau_{y}}^{\prime }=\frac{1}{2}-\frac{1}{2}\;\sqrt{s^{\prime }}$$

The density ρ_m_ was obtained by measuring material weight at constant volume using the same cylindrical mould for all mortar mixture, glass plate, and a digital balance. Mortar was filled in two layers; each layer was compacted with 10 short strokes with the help of a tamping rod to ensure a uniform distribution of material and reduce air pockets in the mould. Extra mortar was scraped off with a trowel, and the overall weight was calculated. Measurement of density is necessary to calculate yield stress and hence was performed for all the mortar mixtures. During the evaluation of buildability of the mortar mixtures via actual printing tests, it was observed that mixtures with high static yield stress exhibited superior shape retention ability and were able to sustain subsequent layers without altering the shape of underlying layers. As a result, high static yield stress can be considered an indicator of enhanced buildability. However, it was also observed that surface cracks tended to occur at high static yield stress values as the material became stiffer. In the case of mixtures 1, 3, 5, 10, and 13, the high static yield stress resulted in unprintable mortar and blockage at the nozzle exit.

#### 2.3.5. The 3D Printing of Mortar

Printing of mortar filaments was carried out with the help of a battery-operated ram extruder with a mortar carrying capacity of 0.6 L. A nozzle with rectangular face is fitted at the front of the ram extruder as shown in Figure 2 below. The nozzle was carefully designed in a rectangular form (internal rectangular geometry 42 × 14 mm) to fit the extruder and with a ribbed interior surface to allow for greater contact area at the surface between subsequent layers. Three layers of mortar filaments, each 200 mm long, can be easily printed with a single filling.

Printing tests were conducted on each trial mix to assess extrudability and buildability. However, there is currently no standardised method for evaluating the extrudability and buildability of a 3D printable mixture. Typically, these characteristics are evaluated through visual inspection, considering factors such as the proper extrusion of the layer(s) without blockage or fracture, the presence of surface fissure or voids during extrusion, the retention of shape or shape stability, and the layer height after printing. Extrudability refers to the ability of concrete to flow through nozzles and is influenced by workability (consistency) and mix proportions [12], while workability refers to the ease of flow in a system and can be measured by the slump flow test [51]. Buildability is commonly evaluated by measuring the height of filament layers, considering the visual deformation or collapse of the structure during 3D printing [11,21,52] and can be associated with static yield stress of the material [9]. The results from the flow table, cone penetration test, and slump test offer comprehensive insights into the extrudability and buildability of the mortar mixture, as discussed in Section 3.1.

#### 2.3.6. Compressive and Flexural Strength Test

For testing compressive strength (***f*′*_cc_***) of mould-cast samples, standard cubes of 50 mm were prepared for testing at 7 days and 28 days. Moulds were filled in two layers compacting each layer with 10 strokes and excess of mortar was removed with a trowel. Printed layered cubes were used to evaluate the compressive strength of layers (***f*′*_cL_***) that were neither vibrated nor compacted. For preparing a printed cube member, three layers were printed in a stacked manner and then cut with a tin cubic mould of 42 mm sides, as seen in Figure 3a. The tin mould was removed one minute later. Since the top surface of layered cubes was not exactly parallel, a cement paste was applied to the cube’s top surface to create parallel surfaces for testing. The preparation of parallel surface for printed cube members with the help of levelling pole can be seen in Figure 3b. This additional material is intended to remedy any residual parallel surface flaws. However, this material is not included in the compressive strength (***f*′*_c_***) calculation, which is shown below in Equation (7). After 24 h of curing, the moulds of the cast samples were removed. All the samples were then cured in water at (20 ± 1) °C for 7 and 28 days, respectively. Both mould-cast and printed samples were tested at a constant load rate of 50 kN/min which conforms to BS EN 1015-11:2019 [53].
(7)$$f_{c}^{\prime }=\frac{Force\;\left[F\right]}{Surface\;area\;\left[A\right]}$$

For testing flexural strength, mould cast samples of size 50 × 50 × 200 mm were prepared while the extruded samples were made of three extruded layers, as shown in Figure 3a, where each layer was approximately 42 mm wide, 14 mm thick, and 180 mm long. However, due to load of subsequent layers placed on top, the depth and width of these printed members may change depending on the flowability and yield stress value of the mortar mixture. Since each printed specimen is different in size, the width (b) and depth (d) of all samples were measured at three different places to obtain an average value for measurement along the plane of failure. According to BS EN 1015-11:2019 [53], flexural strength was calculated by dividing the maximum load the specimen could sustain by its cross-sectional area. Unlike prismatically mould-cast members, the printed specimen is uneven and requires a more accurate result. To address this, three printed member samples made from the same mixture were produced. The width and depth of each specimen across the plane of failure must be measured an average of three times to determine the flexural strength of printed members. The width of each specimen is determined by averaging three measurements taken in the middle of each layer along the sample’s depth. Similarly, the depth is determined by averaging three vertical measurements across the width of the failure plane taken perpendicular to the printing direction. The length of each specimen, measured from one support to the next, was fixed for both printed and mould-cast specimens. The flexural strength of both layered and standard mould-cast prism samples was calculated using the Equation (8) given below, where ***f′_t_*** is the flexural strength, ‘F’ is the maximum flexural force applied before failure, ‘b’ is the width of the sample, ‘d’ is the depth, and ‘l’ the length in between the two supports (in all cases, l = 110 mm):(8)$$f_{t}^{\prime }=1.5\times \frac{F\times l}{b\times d^{2}}$$

After keeping the samples in 100% humid condition and 20 ± 1 °C for 24 h, samples were cured in water for 7 and 28 days before the flexural strength test. The flexural test was a three-point bending test of hardened mortar samples (for both mould-cast and printed members) at 7 and 28 days. In accordance with BS EN 1015-11:2019 [53], samples were tested at a steady load rate of 40 N/s.

## 3. Results and Discussion

### 3.1. Fresh and Hardened Responses of Investigated Mixes

The mix design sought to optimise 3D concrete printing performance both at the fresh and hardened states. Fresh state characteristics such as extrudability and buildability are measured in terms of flowability, consistency and static yield stress. Compressive and flexural strength are investigated to assess the performance at hardened state. A total of twenty mortar mixtures were investigated, and eleven properties were considered for which regression equations were derived using the factorial design approach. Table 5 shows the responses of the 20 mortar mixes in terms of slump flow, cone penetration, yield stress, and compressive and flexural strength at 7 and 28 days. Mixtures 17 to 19 were used to evaluate the repeatability of mortar compositions and are listed in Table 6 below. In this study, measurement of the flexural and compressive strength of 3D printed mortar elements were conducted perpendicular to the direction of printing. Flexural strength at 7 days for mould-cast and printed samples are indicated by ***f′_tc_*_7_** and ***f′_tL_*_7_**, respectively. Similarly, 28-day compressive strengths for mould-cast and printed samples are indicated as ***f′_cc_*_28_** and ***f′_cL_*_28_**, respectively. Since the mortar was not printable for mixes 1, 3, 5, 10, and 13, the printed elements for those mixes are labelled as NP or Not Printable in Table 5.

The extrudability and buildability of a cementitious mortar were evaluated using a battery-operated extruder. Extrudability is a measure of the material’s ability to flow smoothly and continuously through a nozzle [2,12,21], and it is quantified using slump flow and cone penetration. Buildability refers to the mortar’s ability to support the weight of successive layers without yielding due to its weight or nonuniform geometry. It is determined by the material’s static yield stress, which can be estimated from the slump value [10,54]. In this study, the quality of extrudability was evaluated based on the absence of surface voids or fissures and smooth, continuous extrusion. On the other hand, the buildability of the material was considered to be satisfactory if it could support the weight of at least three layers printed without bottom layer collapse, with the height of the bottom layer being equal to that of the top layer. There is ample evidence in the literature [10,12,55] to support that a cementitious mortar with a high static yield stress is more likely to be able to support successive layers and maintain its shape during 3D printing. Mixtures 1, 3, 5, 10, and 13 possess high static yield stress values (>1600 Pa), but they exhibit poor extrudability, as indicated by their low slump flow value of less than 200 mm. The high stiffness and low flowability of these mixtures led to blockages at the nozzle exit, making them unsuitable for 3D printing. The mortar mixtures 2, 4, 6, 8, 9, 11, and 12 have high slump values and penetration depth, making them easily extrudable. However, these mixtures lack buildability as their static yield stress values are insufficient to resist gravity or support subsequent layers. Mixtures 7 and 16 are extruded with surface fissures, but they are buildable due to their ability to retain shape and high yield stress values. The remaining mixtures 14, 15, 17, 18, 19, and 20 can be extruded without any surface deformations and are buildable as they can support subsequent layers with minimal deformations. The results from the flow table, cone penetration, and slump test, combined with findings from the printing tests, as presented in Table 5 and Figure 4, provide a comprehensive understanding of the extrudability and buildability of the mortar mixtures.

Figure 4 is a diagram matrix illustrating the relationship between slump flow and extrusion, as well as slump and buildability, for mixes 2, 9, 16, and 19 from Table 5. Rows 1 to 4 in Figure 4 represent the slump flow, extrudability, yield stress, and buildability, respectively. The slump flow value (295 mm) for mix 2 is very high, making the material easier to extrude. Moreover, the slump for mix 2 is also high, resulting in the estimated yield stress value being low (520 Pa). A mortar with a low yield stress value cannot support subsequent layers on top of it, so that the bottom layers yield under gravity. The slump flow and yield stress computed for mix 16 are 212 mm and 1506 Pa, respectively. The extruded mortar filaments for mix 16 have certain surface flaws (visible cracks and pores) as it is comparatively harder to extrude due to its low slump flow value. However, the mortar layers can sustain a load of subsequent layers while maintaining the shape with minimum settlement as compared to mixes 2, 9, and 19. The quality of extruded mortar layers for mix 19 is somewhat satisfactory, although the shape of those filaments is not better than for mix 16. It is observed that extrusion of mortar becomes easier at higher slump flow values, but the buildability decreases due to its low yield stress.

In summary, the slump flow and estimated yield stress values can be used to assess the extrusion behaviour and buildability of cementitious mortar, which are important qualities for successful 3D printing. These values provide a clear indication of the favourable characteristics for 3D printing mortar.

Figure 4 presents the printed layers alongside the slump and slump flow values for four different mortar mixes, providing a general understanding of the distinction between extrudable and buildable mixtures and highlighting the importance of both slump and slump flow in determining the suitability of a mortar for 3D printing. The flowability and consistency of a mortar, which determine its ability to be shaped by gravity, can be assessed using either the slump flow or cone penetration test. The results in Table 5 show that, as the slump flow value increases, the penetration depth also increases. Figure 5a illustrates the correlation between slump flow and penetration for all of the mixes. Additionally, analysis of the rheological properties reveals a relationship between slump flow and estimated yield stress, with lower slump flow values corresponding to higher estimated yield stress values, as shown in Figure 5b.

The compressive and flexural strengths of the extruded samples are determined through statistical analysis of the available data. To ensure that the results are based on reliable and valid data, only the printable samples are included in the calculation of the main and interaction effects. The factorial design involves fitting a statistical model to the data and estimating the coefficients of the independent variables, as well as the interaction terms between the independent variables in the model. These coefficients reflect the main effects and interaction effects of the independent variables on the compressive and flexural strengths of the extruded samples. In conducting analysis for compressive and flexural strengths of mortars, “Design Expert” software has two options: ignore the missing data (non-printable mixtures 1, 3, 5, 10, and 13) and perform the calculations based on only the available data, or use an imputation method to estimate the missing values. In this study, the first approach is used because the imputation method to estimate missing values can potentially impact the validity and reliability of the results. Imputation methods rely on making assumptions about the missing data and the relationships between the variables, and if these assumptions are not met, the imputed values may not accurately represent the true values of the missing data. Additionally, imputation methods can introduce variability and uncertainty into the results, which may lead to a skewed or biased estimate of the main and interaction effects. This can be concerning if the proportion of missing data is substantial, as it can have a substantial impact on the results of a factorial design study.

The compressive strength values of elements cast in a mould at 7 and 28 days are generally higher than those of 3D printed elements made from the same mixtures, as seen in the test results in Table 5. This improved mechanical performance may be due to the compaction of elements cast in a mould, while 3D printed elements are simply stacked on top of one another. The evolution of compressive strength in standard mould-cast cubes and 3D-printed layered cubes generally follows the same linear trend, yielding a correlation with R^2^ values of 0.825 and 0.822 at 7 and 28 days, respectively, as shown in Figure 6a,b.

The flexural strength values of 3D printed elements are generally higher than those of elements cast in a mould. This behaviour is likely due to the alignment of fibres along the print path [35], facilitated by extrusion through the nozzle, which increases the flexural strength of the printed elements perpendicular to the direction of printing. Both mould-cast prism elements and their 3D-printed layered equivalents exhibit similar trends in flexural strength growth, with R^2^ values of 0.87 and 0.79 at 7 and 28 days, respectively, as shown in Figure 7a,b.

### 3.2. Regression Models and Isoresponse

The data presented in Table 5 obtained using a factorial design method enable the quantification of the individual and interaction effects of the four mixture parameters (CEM, FA, BA, and SP) on the investigated responses. Analysis of variance (ANOVA) was carried out to test the statistical significance of main and interaction effects using the commercial software “Design Expert”. Based on this analysis regression models were then created under the assumption of a normal distribution of residual terms [44]. All model parameters were statistically significant at 95%. This level of confidence is normally considered strong evidence that the parameter is not zero, i.e., that the proposed parameter has a significant influence on the measured response. A positive/negative estimate of the coefficient means that an increase in the given parameter results in an increment/reduction respectively of the measured response. Based on the statistical regression models, isoresponse curves throughout the experimental domain are then created using response surface methodology (RSM).

The regression models built using the factorial design of experiment method were used to predict the values of mixes 17–20, as shown in Table 7. Table 8 lists the ratios of predicted to measured responses for all the fresh state and hardened properties including slump flow, cone penetration, yield stress, ***f′_tc_*_7_**, ***f′_tL_*_7_**, ***f′_tc_*_28_**, ***f′_tL_*_28_**, ***f′_cc_*_7_**, ***f′_cL_*_7_**, ***f′_cc_*_28_**, and ***f′_cL_*_28_**. These ratios range from 0.90 to 1.10, indicating that the developed regression models predict the properties of the selected mortar mixes with reasonable accuracy, with the exception of mix 3 for ***f′_tc_*_7_** (predicted-to-measured = 0.88) and mix 4 for ***f′_cc_*_7_** (predicted-to-measured = 0.89). Overall, the proposed models seem to be satisfactory in predicting the flowability, consistency, static yield stress, flexural strength, and compressive strength of both mould-cast and printed samples at 7 and 28 days.

The developed prediction models can be used to analyse the influence of input parameters on the rheology and strength of the designed mortar mixture. The regression coefficients presented in Table 9 were determined to be statistically significant at 95% confidence level (*p*-value < 0.05). These regression coefficients indicate the strength and direction of the relationship between the investigated parameters and the measured response. A positive coefficient indicates that, as the parameter increases, the response also tends to increase, while a negative coefficient indicates that, as the parameter increases, the response variable tends to decrease. The magnitude of the coefficient represents the strength of the relationship: the larger the coefficient, the stronger the relationship.

#### 3.2.1. Slump Flow

The regression model of slump flow is primarily affected by SP, CEM, and FA concentrations, as well as by the interaction effects of SP, CEM, and FA (R^2^ = 0.90), see Equation (9).
(9)$$Slump\;Flow\left[\mathrm{mm}\right]=226.3+21.9\;CEM+10.2\;FA+29.4\;SP-12.1\;CEM\times FA-20.8\;CEM\times SP-10.6\;FA\times SP$$

Because of its ability to act as a dispersant via electrostatic or steric repulsion effects [50], the content of SP (+29.4) is the most important factor in increasing slump flow followed by CEM (+21.9), see Equation (9). The use of PCE type superplasticiser is known to improve the flowability of fresh concrete [31]. The mixtures with the lowest superplasticiser content (2 kg/m^3^) were noticeably stiffer, making mixtures 1, 3, 5, 10, and 13 not printable. As shown by past research [12,33,56], slump flow can also be enhanced by increasing the dosages of CEM and FA in the mixture. Despite having the lowest SP dosages, mixes 7, 12, and 14 were printable and had slump flow values greater than 200 mm due to the high concentration of CEM and FA [57,58]. The effect of SP on the slump flow of fresh mortar mixes containing CEM and FA is illustrated in Figure 8a,b, respectively. Figure 8a shows the isoresponse curve of the slump flow with a fixed proportion of FA at 12.5% and BA at 2 kg/m^3^. The gradient of the contours represents the significance of each parameter in effecting the slump flow value. When SP dosage was 2.5 kg/m^3^ and CEM content was 570 kg/m^3^, the isoresponse of the predicted slump flow was 190 mm which is difficult to extrude, as observed from mix 3 in Table 5. If SP dosage was increased to 3 kg/m^3^, while maintaining CEM content at 570 kg/m^3^, the isoresponse of the slump value was approximately 210 mm, making the mortar extrudable. Figure 8b presents the isoresponse of slump flow of SP vs. FA content when CEM was fixed at 600 kg/m^3^ and BA at 2 kg/m^3^. In this case the significance of the effect of FA content on slump flow seemed to be less compared to SP. This could be because of the lower percentage of the FA content (5–20%) in the mix composition as compared to the CEM content. From the isoresponse curve it can be inferred that the SP dosage plays a significant role in improving slump flow of the mortar even with a smaller dosage, as compared to CEM and FA.

#### 3.2.2. Cone Penetration

The regression model for the cone penetration is influenced in order of magnitude by SP (+8.5), CEM (+5.6), and FA (+4.0) as presented in Equation (10) (R^2^ = 0.97).
(10)$$Cone\;Penetration\left[\mathrm{mm}\right]=26.1+5.6\;CEM+4.0\;FA+8.5\;SP-4.8\;CEM\times SP-3.8\;FA\times SP$$

The presence of high SP dosages has the most significant impact on the consistency of fresh mortar. Penetration depth increases with increasing SP dosages because the mortar becomes less viscous and therefore flows more easily. The mixes that have high dosages of superplasticiser (4 kg/m^3^) have a noticeably lower viscosity, which makes mixtures 2, 4, 6, 8, 11, and 14 very suitable for extrusion. On the other hand, the mixtures 1, 3, 5, 10, and 13 are not extrudable as the penetration depth obtained for these mixtures was less than 15 mm. To be suitable for 3D printing, the mortar should have a penetration depth of at least 20 mm. Figure 9a,b shows the variations in the SP dosage with both CEM and FA content. The isoresponse contours depicted in Figure 9a,b shows that the influence of SP in improving the consistency of the mortar is more significant than the influence of CEM content or FA content. Furthermore, Figure 9a,b shows that the consistency of mortar becomes more pronounced when both CEM and FA concentrations are increased, potentially due to the sensitivity of the mortar composition to the changes in SP dosage. Maintaining FA constant at 12.5%, CEM at 600 kg/m^3^, and BA at 2 kg/m^3^, the penetration depth increased significantly with increasing SP content from 2 to 4 kg/m^3^, as demonstrated in Figure 9b.

#### 3.2.3. Yield Stress

The regression model of yield stress is primarily affected by the SP (−383.9), FA (+53.8), and BA (+41.3) dosages, as indicated in Equation (11) (R^2^ = 0.99).
(11)$$Yield\;stress\left[\mathrm{Pa}\right]=1122.6+53.8\;FA+41.3\;BA-383.9\;SP+138.6\;CEM\times SP+185.5\;FA\times SP$$

The enhancement of static yield stress values is also contributed by the interaction between FA and SP, with a regression coefficient of +185.5, and CEM and SP, with a regression coefficient of +138.6. Lowering the SP dosage or increasing the FA and BA content results in higher yield stress values, as shown in Equation (11), which also corresponds to a decrease in slump flow values. The increase in SP (−383.9) dosage has nearly seven times greater impact on reducing yield stress than the dosage of FA has on increasing yield stress (+53.8). Adding SP improves the surface coating of cement (CEM) and fly ash (FA) particles by a polymer, resulting in steric hindrance and electrostatic repulsion between the particles, and hence requiring less force to disperse the particles and resulting in lower yield stress [59]. Mortar mixes that include fly ash have been found to offer increased resistance to segregation and improved cohesiveness [33,60], while basalt fibre decreases both the mortar’s fresh flow and its deformation ability [12,33,34]. The yield stress of mortar increases with the incorporation of fly ash and basalt fibre, but the improvement is not as significant as the decrease in yield stress due to the addition of SP, see Equation (11). This behaviour is more likely to occur when the SP dosage applied in the mixture is less than the maximum amount (i.e., 4 kg/m^3^), leaving more space for the FA and BA to act as a barrier to free flow and hence yield stress increase is more significant at low SP dosages as in mix 3, 5, 7, and 13 in Table 2. Therefore, addition of more FA and BA content is able to increase the buildability of mix composition when SP dosage is lower than 4 kg/m^3^. The regression model for yield stress indicates that the interaction between SP and CEM, as well as between FA and SP, has a positive effect on the overall yield stress of the mortar. Figure 10a shows the isoresponse curve of the yield stress with a fixed concentration of CEM at 600 kg/m^3^ and SP at 3 kg/m^3^. The effect of FA content is almost as significant as the effect of BA content on improving the yield stress of 3DPM. An increase in the concentration of FA has approximately 1.3 times more influence on yield stress compared to an increase in the dosage of BA. The improvement in yield stress due to addition of FA content can be attributed to the increase in cohesion and decrease in segregation [61,62]. The basalt fibre acts as reinforcement, distributing the applied stress over a larger volume and preventing the propagation of cracks while improving cohesiveness through the formation of a network in the mortar matrix [33,63]. However, compared to the negative effect of SP (−383.9), the influence of FA (+53.8) and BA (+41.3) content is very small. Figure 10c depicts the isoresponse contours, which describe the interaction effect of FA and SP. At a lower FA%, yield stress values change more rapidly under the influence of the SP dosage. When FA dosage is 8%, the corresponding yield stress at SP dosages of 2 kg/m^3^ and 3.5 kg/m^3^ is 1600 Pa and 800 Pa, respectively, while at FA dosage of 17%, the isoresponse of the predicted yield value is approximately 1430 Pa and 950 Pa for SP dosages of 2 kg/m^3^ and 3.5 kg/m^3^, respectively. Figure 10b illustrates the isoresponse curve of the yield stress in relation to CEM and SP, with a fixed concentration of FA at 12.5% and BA at 2 kg/m^3^. As observed before for the interaction of FA and SP, with the increase in SP dosage in the mortar, the yield stress values decrease more at lower CEM concentrations than they do at higher CEM concentrations. Although the influence of cement on yield stress is not evident from Equation (11) in this experiment, the interaction of CEM and SP has a positive effect on overall yield stress increase, which could be due to SP consumption enabling adequate mixing of mortar composition and inherent behaviour exhibited by CEM such as structuration, thixotropy, and nucleation immediately after mixing with water [13,64]. According to the isoresponse behaviour, the limited effect of CEM content on static yield stress at lower concentrations and higher SP dosages could be due to the superplasticiser’s ability to keep the particles sterically separated, delaying microstructure growth and inhibiting early age hydration [65].

#### 3.2.4. Flexural Strength

Equations (12)–(15) show the regression equation between flexural strengths (***f′_tc_*_7_*, f′_tc_*_28_*, f′_tL_*_7_*,*** and ***f′_tL_*_28_**) and primary parameters (CEM, FA, SP, and BA dosages) at 7 and 28 days (R^2^ = 0.88, 0.82, 0.86, and 0.93), respectively. The flexural strengths of mould-cast members are represented by, ***f′_tc_*_7_** and ***f′_tc_*_28_**, respectively, and the flexural strengths of printed elements are represented by ***f′_tL_*_7_** and ***f′_tL_*_28_**.
(12)$${f^{\prime }}_{tc7}\left[\mathrm{MPa}\right]=7.9+0.3\;CEM+1.1\;SP-0.6\;CEM\times SP$$
(13)$${f^{\prime }}_{tc28}\left[\mathrm{MPa}\right]=9.4+0.6\;CEM+1.1\;SP-0.7\;CEM\times SP$$
(14)$${f^{\prime }}_{tL7}\left[\mathrm{MPa}\right]=9.2-0.4\;CEM+0.5\;BA-0.4\;FA\times BA\;$$
(15)$${f^{\prime }}_{tL28}\left[\mathrm{MPa}\right]=11.5-0.2\;CEM-0.3\;FA+0.5\;BA-0.2\;CEM\times FA+0.3\;CEM\times BA\;$$

As shown in Equations (12) and (13), flexural strength is influenced, in order of magnitude, by the dosages of SP and CEM at both 7 and 28 days. As the values of CEM are increased from 550 kg/m^3^ to 650 kg/m^3^ and dosage of SP is increased from 2 kg/m^3^ to 4 kg/m^3^, each contour demonstrates a noticeable trend, indicating that both of these factors have a significant impact on the flexural strength. The primary purpose of incorporating basalt fibre (BA) is to enhance the flexural strength of mortar through fibre reinforcement. However, the results do not indicate any significant impact of the dosage of basalt fibre (BA) on the 7-day and 28-day flexural strength (***f′_tc_*_7_** and ***f′_tc_*_28_**) of the mould-cast specimens. This could be due to the compaction of the mortar in the mould-cast specimens using a tamping rod, which eliminates the potential alignment along the length of the specimen, preventing it from serving as reinforcement when a force is applied perpendicular to the specimen. This compaction effectively reduces the influence of the basalt fibre (BA) on the flexural strength of mould-cast specimens.

Flexural strength at 7 days for cast specimens: The increase in SP affects 7-day flexural strength approximately 3.7 times more than the increase in CEM content (1.1 vs. 0.3 in Equation (12)). Figure 11a shows that the significance of the gradient of isoresponse for SP dosage is much higher than the CEM content. As shown in Table 4 and Table 5, mortar mixtures 1, 5, 10, and 13 have the lowest SP dosage, resulting in lower 7-day flexural strength in comparison to other mixtures. It is possible that lower SP dosage could lead to improper mixing of binder and fine aggregates because of insufficient cohesion between the components during casting. With low SP dosage, the binder might not adhere properly to the fine aggregates, leading to inadequate mixing and weaker mortar. Despite the low SP dosage, the mixtures 3, 7, 12, and 14 have sufficient flexural strength due to the high CEM content. This increased cement content could lead to a denser concentration of hydration products and a greater degree of interlocking between the cement and aggregates, both of which enhance the flexural strength. The 7-day flexural strength of mixture 15 is 1.7 times stronger than that of mixture 13, which is due to a 2 kg/m^3^ increase in SP dosage, whereas mixture 7 has a 1.4-times increase in 7-day flexural strength, which is attributed to a 100 kg/m^3^ increase in CEM content. For SP dosage less than 3.5 kg/m^3^, increasing the CEM content has a positive effect on ***f′_tc_*_7_**. However, for SP dosages between 3.5–4 kg/m^3^, increasing the CEM dosage appears to decrease the 7-day flexural strength of mould-cast specimens (***f′_tc_*_7_**). Additionally, the interaction of CEM and SP has a negative impact on the overall flexural strength development of (***f′_tc_*_7_**).

Flexural strength at 28 days for cast specimens: The 28-day flexural strength is primarily affected by the SP and CEM content of the mortar mixture. From the regression equation (Equation (13)), it is observed that the increase in SP is 1.8 times more effective in increasing 28-day flexural strength than the CEM content. The gradient of isoresponse for SP dosage is more significant than the gradient of isoresponse for CEM content, as evidenced by the rapid change in strength level with change in SP dosage when compared to the CEM content. The trend in the 28-day flexural strength mirrors that of the 7-day flexural strength. Mixture 4 exhibited the highest flexural strength, with CEM and SP dosages at the upper end of the experimental range, at 650 kg/m^3^ and 4 kg/m^3^ respectively, along with FA and BA dosages of 5% and 3 kg/m^3^, respectively. Figure 11b shows the isoresponse curves for CEM versus SP at 28 days with FA held constant at 12.5% and BA at 2 kg/m^3^. After 28 days, at a fixed SP dosage of 2 kg/m^3^, increasing CEM improves the ***f′_tc_*_28_** by approximately 2.2 MPa (9.5–7.3 MPa), but for a higher SP dosage of 3.5 kg/m^3^, the increase in ***f′_tc_*_28_** is approximately 0.5 MPa (10–9.5 MPa). Figure 11b shows that increasing the dosage of SP in conjunction with an increase in CEM improves the ***f′_tc_*_28_** value. The interaction between CEM and SP for ***f′_tc_*_28_** (regression coefficient = −0.7) is comparable to that of ***f′_tc_*_7_** (regression coefficient = −0.6).

Flexural strength of printed elements at 7-day: As depicted by the regression models represented in Equations (14) and (15), the primary parameters affecting the 7-day flexural strengths (***f′_tL_*_7_**) of printed elements, in order of significance, are the dosages of BA and CEM (+0.5 and −0.4, respectively). Increasing the dosage of BA from 1 to 3 kg/m^3^ enhances the ***f′_tL_*_7_** from approximately 8.5 MPa to 9.7 MPa. The increase in flexural strength of the material being printed can be attributed to the alignment of fibres in the direction of the printing. This alignment is a result of the flow of material becoming more constricted or compressed as it passes through the nozzle, which is caused by the shape of the nozzle allowing the fibres to become more tightly packed and oriented in the direction of the flow. This phenomenon has been previously reported in the literature [66,67]. Figure 12a shows the increase in flexural strength with decreasing concentration of CEM and higher dosage of basalt fibre (BA). The impact of the BA dosage on the 7-day flexural strength is the most significant and positive (+0.5), whereas the impact of the CEM content on the 7-day flexural strength is opposite and negative (−0.4). The mixtures 3, 5, 9, 10, 11, 14, and 16 have the maximum basalt fibre; however, only 9, 11, 14, and 16 are printable with flexural strengths of 9.3, 8.9, 8.8, and 10.7, respectively. Of all mixes considered for factorial design, mixture 16 with a minimum cement content of 550 kg/m^3^ and basalt fibre of 3 kg/m^3^ exhibits the maximum flexural strength. Among the mortar formulations, mixtures 2 and 7 have the lowest 7-day flexural strengths. The high extrudability and low yield stress characteristics of mixture 2 result in a cross section with the lowest depth compared to the other mixtures, thus rendering it incapable of supporting subsequent layers effectively. The mortar specimens corresponding to mixture 2 demonstrate the lowest 7-day flexural strength among all the mixtures, owing to a lower resisting force during the flexural test. The combination of low basalt fibre content and high dosages of SP (4 kg/m^3^) and CEM (650 kg/m^3^) in this mixture contributes to its high extrudability and poor buildability, which ultimately results in a cross-section with the least resistance to bending. In comparison, mixture 7, which has the lowest SP dosage (2 kg/m^3^), the lowest basalt fibre content (1 kg/m^3^), and the highest cement content (650 kg/m^3^), exhibits high static yield stress. This high yield stress and low extrudability (flow value) resulted in surface fissures (Figure 4, mixture 16), which indicates insufficient cohesion in the mortar matrix. The lack of proper interlocking of mix composition and the low basalt fibre dosage could contribute to the low flexural strength observed after 7 days. Consequently, it can be inferred that, in addition to the mix proportion, printing parameters such as extrudability and buildability have a significant impact on the flexural strength properties. The regression equation (Equation (14)) predicts that the two-factor interaction of BA and FA is also significant and influences the flexural strength (***f′_tL_*_7_**) of the printed elements at 7 days.

Flexural strength of printed elements at 28-day: The regression model presented in Equation (15) reveals that the 28-day flexural strength of printed specimens is positively influenced by the dosage of BA (+0.5), but negatively impacted by the dosages of cement CEM (−0.2) and FA (−0.3), as indicated by the respective regression coefficients in Equation (15). Figure 12b illustrates the effect of increasing levels of BA and FA dosage on the 28-day flexural strength of printed specimens (***f′_tL_*_28_**). The trend for 28-day flexural strength regarding the BA and CEM content is found to be consistent with the model for the 7-day flexural strength. Increasing BA dosages has 2.5 times and 1.7 times more effect on increasing the flexural strength than CEM and FA content has on decreasing it, respectively. The effect of an increase in the content of FA on the 28-day flexural strength was not as anticipated, similar to the impact of CEM on the 7-day flexural strength, where higher CEM content resulted in poor buildability and made little contribution to the growth of the flexural strength. The positive contribution of FA on flowability may result in an unfavourable effect on flexural strength.. The increase in flowability, caused by the high FA content, leads to poor buildability and results in unstable layers, causing a reduction in the depth of the cross-section and decreased resistance to bending, ultimately resulting in lower flexural strength at 28 days. From Table 5, it is observed that the mixture 8 with high CEM and FA content but low BA content has the lowest 28-day flexural strength of 9.9 MPa and the highest flowability (296 mm) of all mix compositions. The flexural strength of mixture 16 with low CEM and FA content and maximum BA content is 11.7 MPa. The maximum 28-day flexural strength is observed for mixture 4 (12.5 MPa), which has a high CEM and BA content but a low FA content. In this case, the interaction effect of CEM and BA (+0.3, Equation (15)) might have contributed to the overall flexural strength at 28 days. The positive effect of the interaction between CEM and BA (+0.3) is 1.5 times greater than the negative impact of the interaction between CEM and FA (−0.2). Figure 13 depicts printed members with basalt fibre visible from the side view of a tear.

#### 3.2.5. Compressive Strength

The compressive strength of mould-cast and extruded samples at 7 and 28 days can be predicted by the regression models represented in Equations (16)–(19). These equations show the relationship between the primary parameters (CEM, FA, BA, and SP) and the measured compressive strength of the samples. The compressive strength of mould-cast samples is represented by ***f′_cc_*_7_** and ***f′_cc_*_28_**, and the compressive strength of extruded samples is represented by ***f′_cL_*_7_** and ***f′_cL_*_28_**, respectively. The mathematical regression models are represented in coded values of CEM, FA, SP and BA with coefficients of determination (R^2^) for ***f′_cc_*_7_**, ***f′_cc_*_28_**, ***f′_cL_*_7_**, and ***f′_cL_*_28_** being 0.88, 0.88, 0.86, and 0.96 respectively.
(16)$${f^{\prime }}_{cc7}\left[\mathrm{MPa}\right]=41.4+5.4\;CEM+5.8\;SP-2.4\;CEM\times SP$$
(17)$${f^{\prime }}_{cc28}\left[\mathrm{MPa}\right]=50.6+6.1\;CEM+2.0\;FA+5.3\;SP-1.7\;FA\times SP\;$$
(18)$${f^{\prime }}_{cL7}\left[\mathrm{MPa}\right]=39.0+2.8\;CEM+3.7\;SP+2.6\;CEM\times FA\;$$
(19)$${f^{\prime }}_{cL28}\left[\mathrm{MPa}\right]=47.0+4.5\;CEM+1.5\;FA+2.2\;SP-2.5\;FA\times SP\;$$

Compressive strength of mould-cast elements at 7 and 28 days: The compressive strength of mould-cast specimens at 7 and 28 days is significantly influenced by CEM and SP, as indicated by the positive regression coefficients in Equations (16) and (17), respectively. The regression equation indicates that if the dosage of both CEM and SP is increased, the compressive strength increases at 7 and 28 days. An increase in SP dosage has a greater impact on the 7-day compressive strength, whereas an increase in CEM has a greater influence on the 28-day compressive strength. The increase in SP affects 7-day compressive strength approximately 1.1 times more than the increase in CEM content (5.8 vs. 5.4 in Equation (16)). Similarly, the increase in CEM content affects 28-day compressive strength approximately 1.2 times more than the increase in SP dosage (6.1 vs. 5.3 in Equation (17)). FA (+2.0) has a positive effect on the 28-day compressive strength, although to a lesser extent than CEM and SP. This is consistent with the research showing an improvement in the compressive strength of concrete mixed with fly ash (FA) and superplasticiser (SP) [68,69]. Figure 14a,b illustrates isoresponse curves, also showing the primary effect of CEM and SP on 7-day and 28-day compressive strength (***f′_cc_*_7_**, ***f′_cc_*_28_**). The significance of the factors can be judged according to the gradient of the response curve. As the level of parameters increases, the significance of CEM and SP gradually increases for 7-day compressive strength. Similarly, the increase in the levels of CEM and SP illustrates a linear increase in the compressive strength at 28 days. The isoresponse curve indicates a maximum compressive strength of approximately 50 MPa and 60 Mpa for 7-day and 28-day strength, respectively. The increase in fly ash content does not seem to have a significant impact on the 7-day compressive strength, and its influence on the 28-day compressive strength is relatively minor (+2.0). This is likely due to the slower pozzolanic reaction of fly ash at an early age. Mixtures 1, 5, 10, and 13 have low CEM and SP contents, which could lead to improper mixing of the binder and sand because of the insufficient cohesion between the mix components during casting. With a low SP dosage and a lack of cohesion, complete hydration cannot be achieved. Moreover, this could render the material more porous and prone to cracking, leading to reduced strength. Despite the low SP dosage, mixtures 3, 7, 12, and 14 exhibited sufficient compressive strength owing to the high CEM content. This increased cement content could lead to a denser concentration of hydration products and a greater degree of interlocking between the cement and aggregates, both of which enhance compressive strength. Mixtures 4, 8, and 11 have high CEM and SP contents and therefore exhibit their highest compressive strengths at 7 days (47.6, 51.9, and 52.1 MPa, respectively) and 28 days (66.4, 61.2, and 58.2 MPa, respectively).

Compressive strength of printed elements at 7- and 28-day: As demonstrated in Figure 15a, there is a linear increase in ***f′_cL_*_7_** with increasing concentrations of both CEM and SP. The regression equation (Equations (18) and (19)) indicates that if the dosage of both CEM and SP is increased, the compressive strength increases at 7 and 28 days. The primary parameters SP (+3.7) and CEM (+2.8) have a significant influence on the development of the 7-day compressive strength for printed members (***f′_cL_*_7_**). The increase in SP affects 7-day compressive strength approximately 1.3 times more than the increase in CEM content (3.7 vs. 2.8 in Equation (18)). The improvement in compressive strength of the 3D-printed cementitious material is due to the uniform distribution of cement particles within the mortar matrix, which is facilitated by the presence of superplasticiser (SP). The increase in SP dosage helps to homogeneously distribute the cement particles within the mortar, producing a more uniform and denser structure, which in turn results in improved 7-day compressive strength, as has been documented in previous studies [70,71]. The inclusion of CEM in the mixture enhances the 7-day compressive strength owing to the formation of additional hydration products. Although fly ash (FA) does not have a direct impact on the 7-day compressive strength (***f′_cL_*_7_**) of the 3D-printed specimens, the combination of cement and fly ash appears to have a positive effect on the overall 7-day compressive strength (***f′_cL_*_7_**) (as demonstrated by Equation (18)). Unlike the flexural strength test, where the layers are allowed to stretch freely and the specimens are obtained based on the extrudability and buildability of the mortar, the printed specimens for the compressive strength tests are obtained using a thin tin mould that is pushed through the fresh layers. The process of extracting the printed sample does not compromise the compressive strength because the mould is only pushed through the sample once it has achieved sufficient strength to withstand any applied force. This is performed to maintain a rectangular shape of the specimen similar to its mould-cast counterpart, which is required for compression testing (as shown in Figure 3).

The regression Equation (19) implies that increasing the dosages of CEM and SP contributes to the linear increase in the 28-day compressive strength (***f′_cL_*_28_**), as shown in Figure 15b. The contours of the isoresponse for 28-day compressive strength show that the significance of the levels of CEM content is much higher than those of SP dosage. The concentration of CEM (+4.5) is the most important parameter in increasing the overall compressive strength (***f′_cL_*_28_**), followed by the concentration of SP (+2.2) and FA (+1.5). The impact of increasing cement content (CEM) on 28-day compressive strength (***f′_cL_*_28_**) is approximately 2.1 times greater than the impact of increasing the dosage of superplasticiser (SP) and 3 times greater than the impact of increasing the content of fly ash (FA). The effect of CEM content on increasing additional hydration products at 28 days seems to have more significance on 28-day compressive strength (***f′_cL_*_28_**) than producing a uniform distribution of mortar mixture facilitated by SP dosage which is otherwise the case for 7-day compressive strength (***f′_cL_*_7_**). This could be because the influence of CEM content on 7-day compressive strength may not be as significant as on 28-day compressive strength (***f′_cL_*_28_**), as 7 days is still an early stage in the hydration process. During the first 7 days, the CEM might not have added to the strength of the mixture by making more hydration products. Similar to the influence of CEM on 28-day compressive strength (***f′_cL_*_28_**), FA content also makes a contribution towards 28-day compressive strength (***f′_cL_*_28_**) by adding more hydration products [72]. It is highly unlikely that the strength development by FA content could contribute to 7-day compressive strength (***f′_cL_*_7_**) of printed specimens. The interaction of FA and SP is observed to have a negative impact on ***f′_cL_*_28_**. Figure 15c shows the effect of fly ash (FA) and superplasticiser (SP) on the 28-day compressive strength (***f′_cL_*_28_**) of 3D printed members, with a constant cement (CEM) and basalt fibre (BA) dosage of 600 kg/m^3^ and 2 kg/m^3^, respectively. The isoresponse contours in Figure 15c illustrates that the effect of FA content is not as significant as that of SP dosage and, as the level of both FA and SP increases, it results in an increase in 28-day compressive strength (***f′_cL_*_28_**)_._ The increase in compressive strength with higher FA content increases up to about 48.3 MPa with a maximum SP level of approximately 3.5 kg/m^3^, and after that compressive strength (***f′_cL_*_28_**) for level of SP beyond 3.5 kg/m^3^ increases with a decrease in FA content.

### 3.3. Desirability Functions for Optimisation

A multi-objective optimisation of 3DCP is performed using a statistical model and a desirability function. The objective of optimisation is to identify the ideal CEM, FA, BA, and SP dosages in order to simultaneously optimise slump flow, penetration, yield stress, and flexural and compressive strength of 3D printed/mould-cast specimens in order to produce the most suitable mortar for 3D printing.

In the desirability function approach for multi-objective optimisation, each objective Y_i_ is associated with a desirability function D_i_ which varies between 0 and 1. To achieve the desired target value (T) of the objective Y_i_, the design variables must be selected to maximise the desirability function so that D_i_ lies closer to 1. Therefore, the desirability function will be D_i_ ≈ 1 if the objective is on target and Di ≈ 0 if objective is outside the acceptable region.

If the goal is to maximise the objective Y_i_, the desirability function is represented mathematically by Equation (20), where L represents the lower bound of the acceptable region and r is the weight associated with the objective.
(20)$$D_{i}=\left\{\begin{array}{c} 0\\ {\left\{\frac{Y_{i}-L}{T-L}\right\}}^{r}\\ 1 \end{array}\right.\begin{array}{c} Y_{i}<L\\ ;\;\;\;\;\;\;\;L\leq Y_{i}\leq T\\ Y_{i}>T \end{array}$$

If the goal is to minimise the objective Y_i_, the desirability function is represented mathematically by Equation (21), where U represents the upper bound of the acceptable region:(21)$$D_{i}=\left\{\begin{array}{c} 1\\ {\left\{\frac{U-Y_{i}}{U-T}\right\}}^{r}\\ 0 \end{array}\right.\begin{array}{c} Y_{i}<T\;\;\;\;\;\;\;\;\;\;\;\;\;\;\;\;\;\;\;\;\;\;\;\;\;\;\;\;\;\;\;\;\;\;\;\;\;\;\;\;\;\;\;\;\;\;\;\;\;\;\;\;\;\;\;\;\;\;\;\;\;\;\;\;\;\;\;\;\;\;\;\;\;\;\;\;\;\;\;\;\;\;\;\;\;\;\;\;\;\;\;\;\;\;\;\;\;\;\;\\ ;\;\;\;\;\;\;\;T\leq Y_{i}\leq U\\ Y_{i}>U\;\;\;\;\;\;\;\;\;\;\;\;\;\;\;\;\;\;\;\;\;\;\;\;\;\;\;\;\;\;\;\;\;\;\;\;\;\;\;\;\;\;\;\;\;\;\;\;\;\;\;\;\;\;\;\;\;\;\;\;\;\;\;\;\;\;\;\;\;\;\;\;\;\;\;\;\;\;\;\;\;\;\;\;\;\;\;\;\;\;\;\;\;\;\;\;\;\;\; \end{array}$$

If the target lies between the lower (L) and upper (U) bounds, the two-sided desirability function is defined as:(22)$$D_{i}=\left\{\begin{array}{c} 0\\ {\left\{\frac{Y_{i}-L}{T-L}\right\}}^{r_{1}}\\ {\left\{\frac{U-Y_{i}}{U-T}\right\}}^{r_{2}}\\ 0 \end{array}\right.\begin{array}{c} Y_{i}<L\\ \;\;\;\;\;\;\;\;L\leq Y_{i}\leq T\\ ;\;\;\;\;\;\;\;T\leq Y_{i}\leq U\\ Y_{i}>U \end{array}$$

The process of evaluating multiple responses begins by transforming them into a singular metric through the calculation of individual desire scores. This is achieved by individually assessing each response and determining a numerical representation of its desirability, and then maximizing the weighted geometric mean of all of these desire scores. This method effectively consolidates the multi-response problem into a simplified single-response evaluation, by providing a consolidated measure of overall desire score.

If D_1_, D_2_, and D_3_ are individual desire scores of individual responses measured, then the overall desirability [44] (pp. 498–500) is represented as:(23)$$D={\left(D_{1}\times D_{2}\times D_{3}\times \ldots \;\ldots \;\ldots .\times D_{n}\right)}^{\frac{1}{n}}$$

Equations (20)–(22) correspond to the algorithms for achieving maximum, minimum, or target in range values for the desirability scores of individual responses. The desirability D is the result of taking the n^th^ root of the product of power exponents of each D_i_, where n is the number of parameters considered in the optimisation process, as depicted by Equation (23). Multi-objective optimisation is achieved by maximum D to obtain the best combination of parameters.

Table 10 summarises the optimisation goals and values for overall desirability for two different sets of targets (Case 1 and Case 2).

The desirability function aims to find the optimal values for various measured responses, including slump flow, penetration, yield stress, and flexural and compressive strength, for both printed and mould cast specimens at 7 and 28 days. Design Expert software is used to optimise the mix proportion of 3DPM with the mortar mixture in two cases (Case 1 and Case 2) using the Optimisation Numerical module. The optimisation is based on a desirability function and results in a mix proportion with different dosage of parameters and measured responses for each case. For Case 1, the mixture parameters and measured responses are constrained to be within the values of the experimental range, with the exception of slump flow and yield stress, which are set for the maximum performance goal to ensure that the mortar mixture is flowable for ease of extrusion and have high yield stress for better buildability. On the other hand, Case 2 focuses on minimizing the cement content while maximizing the levels of both fly ash and basalt fibre. The optimisation process aims to find the optimal mix composition that is easily extrudable and buildable, while also producing maximum flexural and compressive strengths for printed specimens at 7 and 28 days. The added constraints in goals for parameters and responses likely impacted the mix proportion, resulting in a lower desirability value. The optimal mix ratios for each case are shown in Table 10. The desirability surfaces and their contour lines are shown in Figure 16.

From the results of Case 1 and Case 2, it is observed that the overall desirability value in Case 2 decreases from 0.726 to 0.516, making the mortar mixture less favourable. The increase in fly ash (FA) and superplasticiser (SP) content in Case 2 results in a more extrudable mixture (247 mm vs. 235 mm) but with lower buildability (1088 Pa vs. 1452 Pa) compared to Case 1. Although the same set of requirements are set for slump flow and yield stress in both Case 1 and Case 2, owing to a two-fold increase in SP and 79% increase in fly ash content, the resulting mortar has a higher yield stress and better flowability. The flexural strengths of the printed specimens in both cases are comparable; however, a slight decrease can be attributed to the increased dosage of fly ash (FA) in Case 2. The regression equations (Equations (12) and (13)) demonstrate that the FA content is the likely cause of the reduction in flexural strength, as the other parameters are constant between Case 1 and Case 2. The 7 and 28-day flexural strengths of mould-cast specimens have increased in Case 2, which can be attributed to a two-fold increase in superplasticiser (SP) dosage as observed in Equations (14) and (15). On the other hand, the increase in the compressive strength of both printed and mould-cast specimens at 28 days as well as in the mould-cast samples at 7 days can be observed due to the increase in the SP and FA content for Case 2. The observed increase in compressive strength due to the increase in FA and SP is expected as per the regression equations derived from material behaviour (see Equations (16) to (19)). However, owing to the increased fly ash (FA) content and the negative interaction effect of cement (FA) and basalt fibre (BA), the 7-day flexural strength of the printed specimens also slightly decreased. Similarly, the adverse effect of increase in FA is also observed in 28-day flexural strength for printed specimens.

This comparison highlights the delicate balance between various parameters (CEM, FA, BA, and SP) and goals set in the multi-objective optimisation procedure in accordance with material behaviour predicted by regression models. The comparative analysis between the two cases is also able to show how altering one aspect can impact the ultimate outcomes of the desirability value. Hence, it can be deduced that determining an ideal mixture composition that harmoniously balances all parameters and satisfies the set objectives is a complex task that requires a structured and methodical approach and factorial design method can be used to obtain an optimised mix composition suitable for 3D printing.

## 4. Conclusions

This study provided a novel approach for proportioning 3D printed mortar by varying the Portland cement (CEM), fly ash (FA), superplasticiser (SP), and basalt fibre (BA) dosages based on the desired performance for both rheological and material strength properties. The proposed regression models, developed using the factorial design approach, can be used for a wide range of mix proportioning, and provide an efficient technique for assessing the influence of mix parameters on measured properties. The models also provide a means for optimizing 3DPM mixtures using a desirability value obtained with the above mix components. Based on the results of this paper, the following conclusions can be made:This study employs standardised, field-friendly protocols to assess the extrudability and buildability of a 3D printable mortar through the utilisation of a ram extruder. There is a close relationship between the extrudability and slump flow, or cone penetration, as well as between the buildability and yield stress of this printable mortar. The suitable rheology and satisfactory printing effects were obtained only for specific values of slump flow, cone penetration, and static yield stress. The investigation of hardened properties demonstrated that the specimens produced via mould casting exhibit a greater compressive strength, whereas those fabricated using 3D printing possess a superior flexural strength, both demonstrating similar trends in their respective strengths for all mix compositions.The study investigated the influence of increased levels of cement and superplasticiser on the characteristics of the mix composition. An increase in the dosages led to an enhancement in the extrudability and compressive strength, yet concurrently resulted in a reduction in the buildability. Although flexural strength improved in mould-cast samples, flexural strength in printed samples decreased with increasing cement content.The integration of fly ash into the mortar composition significantly improves the flow value and consistency. Although the addition of fly ash has a positive impact on the yield stress, this effect is countered by the addition of a superplasticiser, which has a much greater impact on the yield stress reduction. Despite having no significant influence on the flexural strength of mould-cast samples, a decline in the 28-day flexural strength of printed samples is observed as the concentration of fly ash increases. However, as the dose of fly ash increases, the compressive strengths of both mould-cast and printed samples increase.The findings of this study suggest that incorporating basalt fibre in the range of 1–3 kg/m^3^ has a positive effect on the static yield stress and flexural strength of the printed elements. The increase in flexural strength is demonstrated by a significant increase in flexural strength at 7 and 28 days when compared to their traditional mould-cast counterparts.The regression models were found to effectively predict the behaviour of the mortar mixture, offering the potential for optimisation of the mix proportion for 3DPM. The use of a desirability function allowed for the identification of an optimal mortar mixture that satisfied the criteria for extrudability, buildability, and hardened strength properties, enabling the simultaneous optimisation of multiple objectives for 3DPM. The validity of the optimised mix proportion is confirmed through further laboratory testing.A major challenge encountered in this study was the manual printing process utilizing a ram extruder. This method presents certain limitations in terms of repeatability and the potential for tilting to occur when printing more than eight layers. The stacking of multiple layers can sometimes result in a slight misalignment, which accumulates and eventually leads to geometric failure. The buildability and compressive and flexural strengths of the samples might also be affected. In an effort to address the limitation regarding repeatability, extra care was exercised during the printing process to minimise surface unevenness and misalignment. This resulted in the achievement of good repeatability, as demonstrated by the outcomes presented in the study. To minimise the effect of tilting, the current study was conducted with a maximum of four layers to ensure reliable and valid results, minimizing the impact of human error. Despite the limitations, the optimisation of mix proportion to obtain a suitable mortar mixture for 3D-printing was carried out and both fresh and hardened properties of 3DPM were successfully investigated.

## Figures and Tables

**Figure 1 materials-16-01748-f001:**
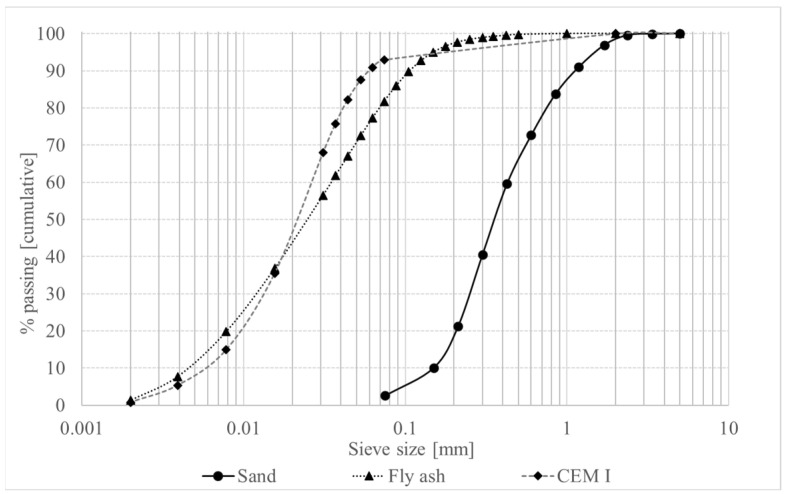
Grain size distribution.

**Figure 2 materials-16-01748-f002:**
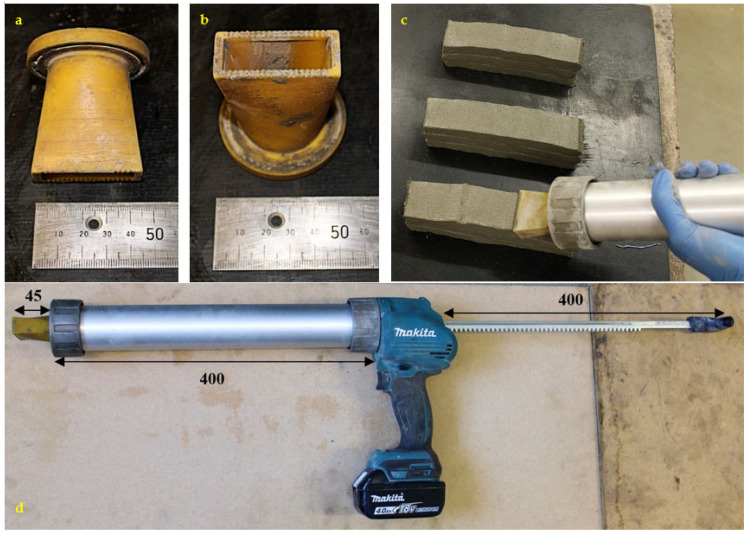
The printing system used for printing of mortar (all dimensions are in mm): (**a**) side view of the nozzle, (**b**) front view of nozzle, (**c**) battery operated ram extruder device fitted with nozzle, and (**d**) a few printed sample later used for flexural strength test of extruded samples.

**Figure 3 materials-16-01748-f003:**
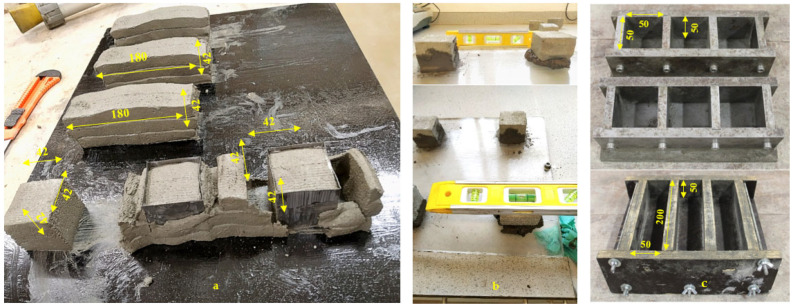
Sample preparation (**a**) for compressive and flexural strength measurement of printed members using the extruder, (**b**) preparing a flat surface for extruded samples for compressive strength test, and (**c**) moulds for preparing cast samples to test compressive and flexural strength.

**Figure 4 materials-16-01748-f004:**
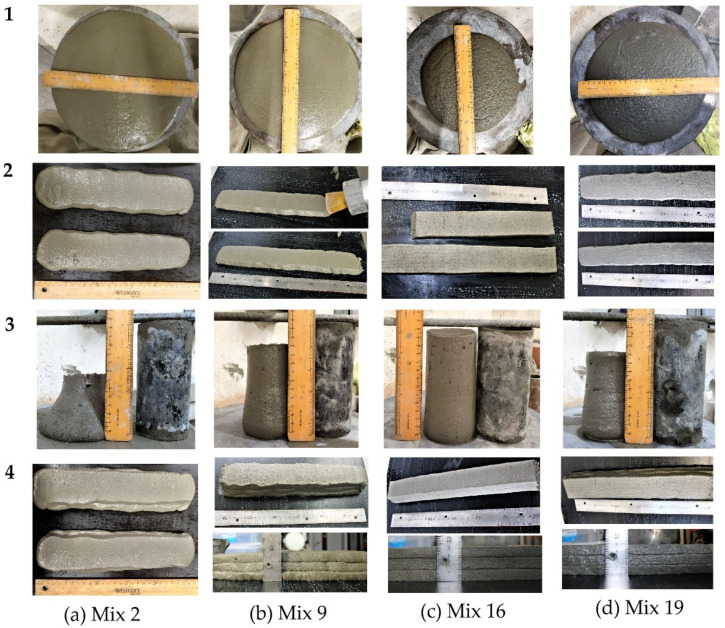
Measurement of slump flow and slump along with printed filaments corresponding to (**a**) mixture no. 2; (**b**) mixture no. 9; (**c**) mixture no. 16; and (**d**) mixture no. 19.

**Figure 5 materials-16-01748-f005:**
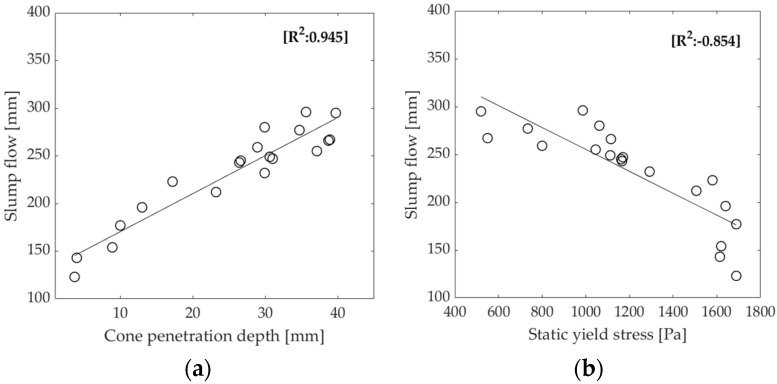
(**a**) Correlation between slump flow and penetration depth, and (**b**) correlation between slump flow and yield stress for all mortar mixes in this experiment (mix 1−20).

**Figure 6 materials-16-01748-f006:**
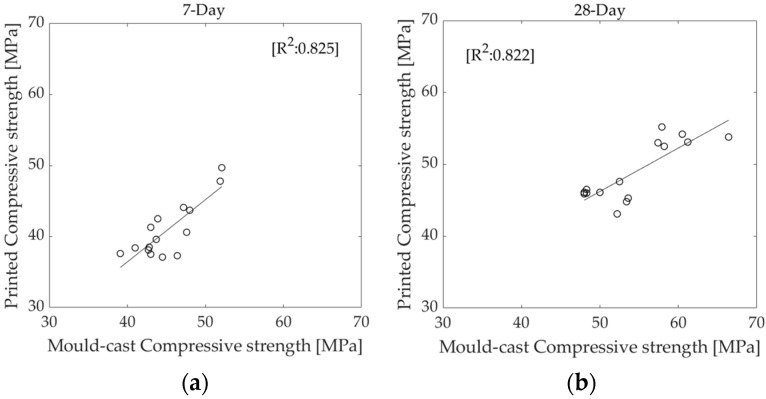
Showing correlation between mould-cast and printed members for (**a**) compressive strength at 7 days and (**b**) compressive strength at 28 days.

**Figure 7 materials-16-01748-f007:**
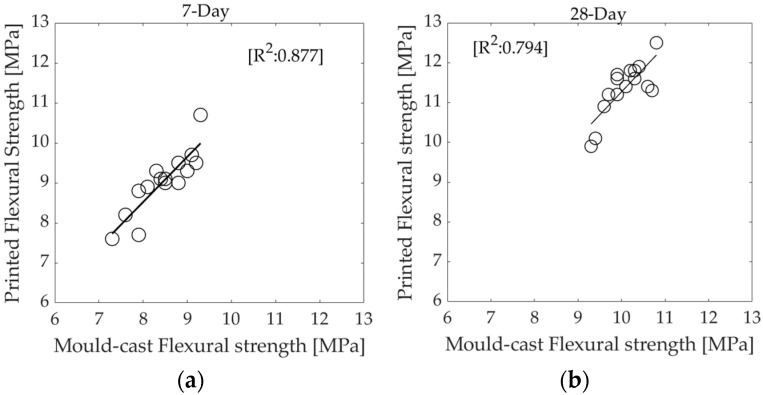
Showing correlation between mould-cast and printed members for (**a**) flexural strength at 7 days, and (**b**) flexural strength at 28 days. Table 6 shows the average measured response of the four replicate mortars mixtures (Mix 17 to 20) prepared at the centre of the experimental domain, coefficient of variation, and half width of 95% confidence interval for each of the measured properties.

**Figure 8 materials-16-01748-f008:**
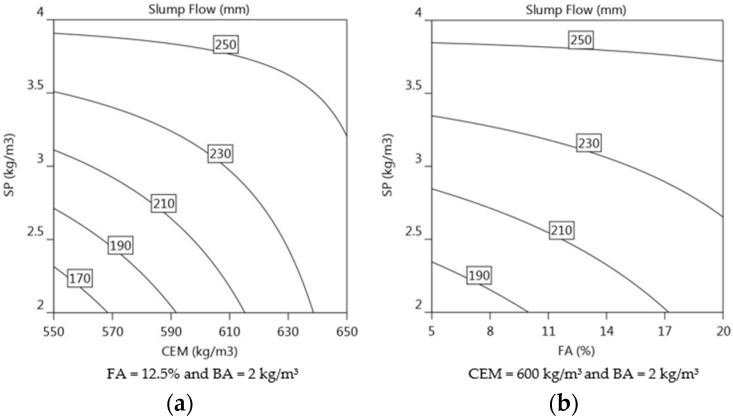
Isoresponse curves of slump flow showing (**a**) SP versus CEM, (**b**) SP versus FA.

**Figure 9 materials-16-01748-f009:**
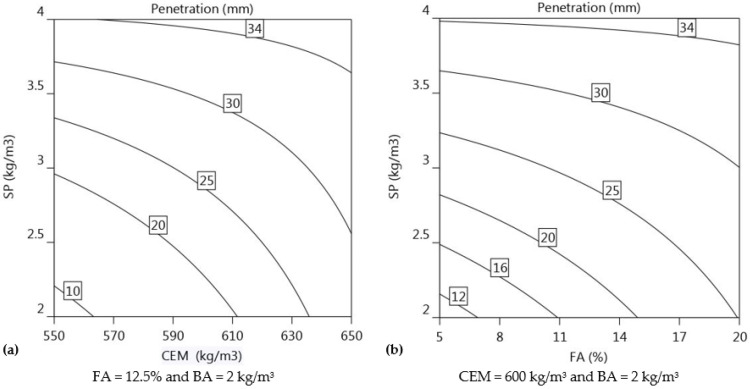
Isoresponse curves of cone penetration test: (**a**) SP versus CEM and (**b**) SP versus FA.

**Figure 10 materials-16-01748-f010:**
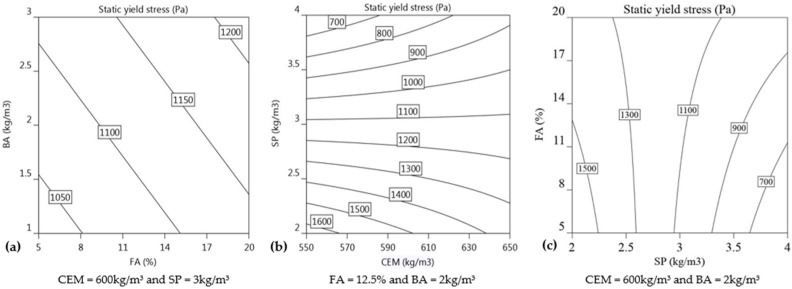
Isoresponse curves illustrating yield stress for (**a**) BA versus FA, (**b**) SP versus CEM, and (**c**) FA versus SP.

**Figure 11 materials-16-01748-f011:**
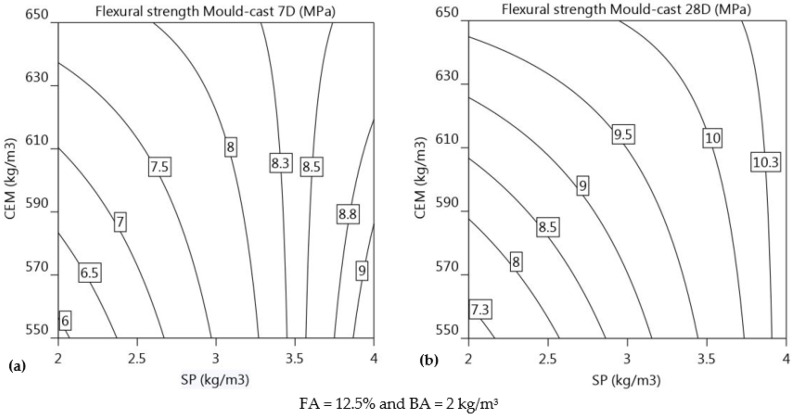
Isoresponse curves for flexural strength of mould-cast members (**a**) CEM versus SP at 7 days, and (**b**) CEM versus SP at 28 days.

**Figure 12 materials-16-01748-f012:**
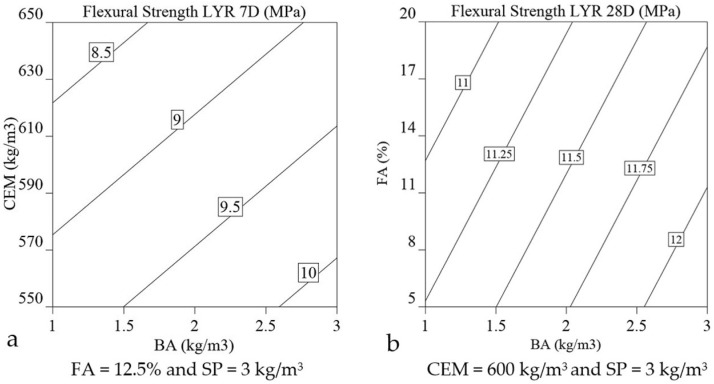
Isoresponse curves showing flexural strength of printed members (**a**) CEM versus BA at 7 days; (**b**) FA versus BA at 28 days.

**Figure 13 materials-16-01748-f013:**
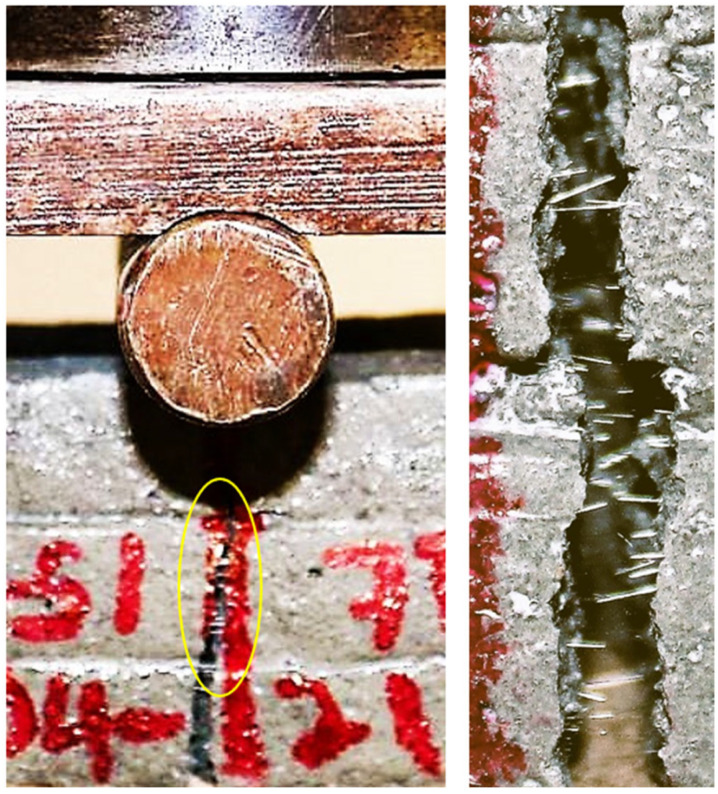
Basalt fibre aligned along the direction of printing and visible through the crack after failure in the 3-point bending test.

**Figure 14 materials-16-01748-f014:**
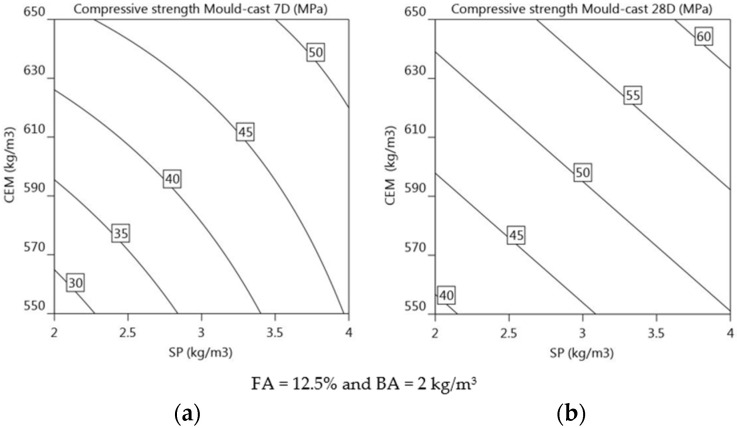
Isoresponse curves for compressive strength of mould-cast members (**a**) CEM versus SP at 7 days; (**b**) CEM versus SP at 28 days.

**Figure 15 materials-16-01748-f015:**
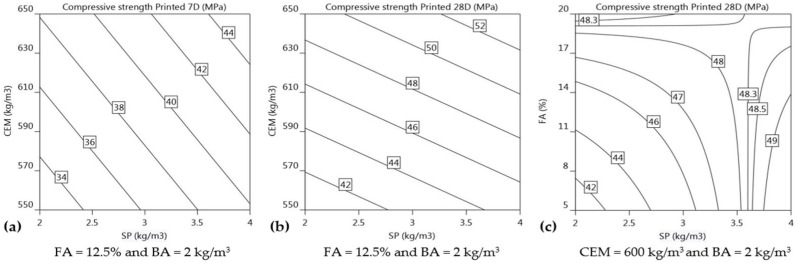
Isoresponse curves for compressive strength of printed members (**a**) CEM versus SP at 7 days, (**b**) CEM versus SP at 28 days, and (**c**) FA versus SP at 28 days.

**Figure 16 materials-16-01748-f016:**
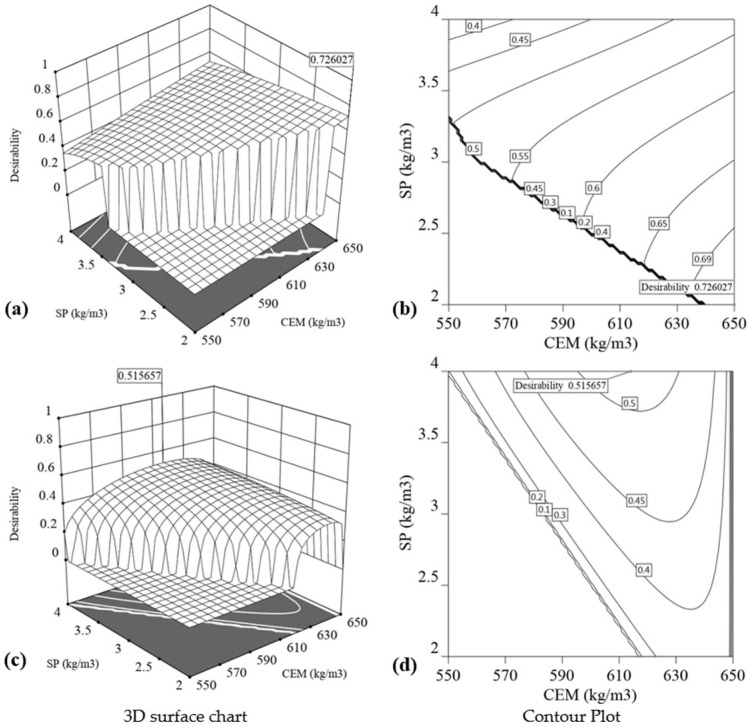
The 3D surface chart and contour plot of desirability obtained for (**a**,**b**) Case 1 and (**c**,**d**) Case 2.

**Table 1 materials-16-01748-t001:** Chemical compositions of cement and fly ash.

Composition [%]	Cement	Fly Ash
SiO_2_	19.83	55.2
Al_2_O_3_	4.87	22.8
Fe_2_O_3_	2.79	6.2
SO_3_	2.51	1.1
MgO	1.98	1.2
CaO	64.62	4.3
Na_2_O	0.19	0.8
K_2_O	0.65	1.9
Cl	0.084	0.1

**Table 2 materials-16-01748-t002:** Physical properties of cement and fly ash.

	Cement	Fly Ash
Loss on ignition (%)	1.67	-
Specific gravity	3.10	2.21
% Passing 45 µm sieve	81	83
Mean particle size (µm)	32	-

**Table 3 materials-16-01748-t003:** Factors and levels considered.

Components	Coded Values		
	$Conc_{low}(-1)$	Centre Points	$Conc_{high}(+1)$
Cement (CEM)	550 kg/m^3^	600 kg/m^3^	650 kg/m^3^
Fly-ash (FA)	5%	12.50%	20%
Superplasticiser (SP)	2 kg/m^3^	3 kg/m^3^	4 kg/m^3^
Basalt fibre (BA)	1 kg/m^3^	2 kg/m^3^	3 kg/m^3^

**Table 4 materials-16-01748-t004:** Mix proportions of mortar (with coded values).

	Mix	Cement [kg/m^3^]	Fly Ash [%]	Basalt Fibre [kg/m^3^]	Superplasticiser [kg/m^3^]
Selected mixtures	1	550 (−1)	20 (+1)	1 (−1)	2 (−1)
	2	650 (+1)	5 (−1)	1 (−1)	4 (+1)
	3	650 (+1)	5 (−1)	3 (+1)	2 (−1)
	4	650 (+1)	5 (−1)	3 (+1)	4 (+1)
	5	550 (−1)	5 (−1)	3 (+1)	2 (−1)
	6	550 (−1)	20 (+1)	1 (−1)	4 (+1)
	7	650 (+1)	5 (−1)	1 (−1)	2 (−1)
	8	650 (+1)	20 (+1)	1 (−1)	4 (+1)
	9	550 (−1)	20 (+1)	3 (+1)	4 (+1)
	10	550 (−1)	20 (+1)	3 (+1)	2 (−1)
	11	650 (+1)	20 (+1)	3 (+1)	4 (+1)
	12	650 (+1)	20 (+1)	1 (−1)	2 (−1)
	13	550 (−1)	5 (−1)	1 (−1)	2 (−1)
	14	650 (+1)	20 (+1)	3 (+1)	2 (−1)
	15	550 (−1)	5 (−1)	1 (−1)	4 (+1)
	16	550 (−1)	5 (−1)	3 (+1)	4 (+1)
Centre point mixtures	17	600	12.5	2	3
	18	600	12.5	2	3
	19	600	12.5	2	3
	20	600	12.5	2	3

**Table 5 materials-16-01748-t005:** Experimental results.

Mix	Slump Flow [mm]	Cone Penetration [mm]	Yield Stress [Pa]	*f′_tc_*_7_ [MPa]	*f′_cc_*_7_ [MPa]	*f′_tL_*_7_ [MPa]	*f′_cL_*_7_ [MPa]	*f′_tc_*_28_ [MPa]	*f′_cc_*_28_ [MPa]	*f′_tL_*_28_ [MPa]	*f′_cL_*_28_ [MPa]
1	154	8.9	1620	5.4	22.9	NP	NP	5.9	40.3	NP	NP
2	295	39.7	520	7.9	48.0	7.7	43.7	10.7	57.9	11.3	55.2
3	196	13.0	1640	7.5	41.6	NP	NP	8.9	43.9	NP	NP
4	267	38.9	550	9.1	47.6	9.7	40.6	10.8	66.4	12.5	53.8
5	123	3.7	1689	5.9	24.1	NP	NP	7.3	32.8	NP	NP
6	277	34.7	734	8.8	42.8	9.5	38.5	10.6	53.4	11.4	44.8
7	223	17.2	1580	7.3	43.0	7.6	37.5	9.6	48.0	10.9	45.9
8	296	35.6	986	8.8	51.9	9.0	47.8	9.3	61.2	9.9	53.1
9	259	28.9	800	9.0	42.7	9.3	38.1	10.4	52.2	11.9	43.1
10	177	10.0	1689	6.2	33.6	NP	NP	7.9	37.4	NP	NP
11	280	29.9	1062	8.1	52.1	8.9	49.7	10.2	58.2	11.8	52.5
12	255	37.1	1045	7.6	43.7	8.2	39.6	9.4	57.4	10.1	53.0
13	143	4.0	1614	5.4	29.2	NP	NP	6.4	41.9	NP	NP
14	266	38.7	1115	7.9	43.9	8.8	42.5	9.9	60.5	11.6	54.2
15	232	29.9	1292	9.2	43.0	9.5	41.3	10.3	50.0	11.6	46.1
16	212	23.2	1506	9.3	47.2	10.7	44.1	9.9	48.3	11.7	46.5
17	243	26.4	1166	8.5	44.5	9.0	37.1	10.3	53.6	11.8	45.3
18	245	26.6	1162	8.3	46.4	9.3	37.3	10.1	48.0	11.4	46.1
19	249	30.6	1112	8.5	41.0	9.1	38.4	9.7	52.5	11.2	47.6
20	247	31.0	1170	8.4	39.1	9.1	37.6	9.9	48.3	11.2	46.0

NP: Not Printable.

**Table 6 materials-16-01748-t006:** Repeatability of test parameters at the central point.

Measured Properties	Slump Flow [mm]	Cone Penetration [mm]	Yield Stress [Pa]	*f′_tc_*_7_ [MPa]	*f′_cc_*_7_ [MPa]	*f′_tL_*_7_ [MPa]	*f′_cL_*_7_ [MPa]	*f′_tc_*_28_ [MPa]	*f′_cc_*_28_ [MPa]	*f′_tL_*_28_ [MPa]	*f′_cL_*_28_ [MPa]
Mean	246.0	28.7	1152.5	8.4	42.8	9.1	37.6	10.0	50.6	11.4	46.3
Std. Dev.	2.6	2.5	27.2	0.1	3.3	0.1	0.6	0.3	2.9	0.3	1.0
Coefficient of variance (%)	1.0	8.7	2.4	1.1	7.7	1.4	1.5	2.6	5.7	2.5	2.1
Half width of 95% confidence interval	4.1	4.0	43.3	0.2	5.3	0.2	0.9	0.4	4.6	0.5	1.5

**Table 7 materials-16-01748-t007:** Simulated mix composition of mortar.

Simulated Mixture Component	Mix 1	Mix 2	Mix 3	Mix 4
Cement (CEM) [kg/m^3^]	620	600	615	635
Fly-ash (FA) [%]	20	20	10	15
Superplasticiser (SP) [kg/m^3^]	3.5	3	2.5	4
Basalt fibre (BA) [kg/m^3^]	3	3	2.5	1.5

**Table 8 materials-16-01748-t008:** Ratio of predicted-to-measured values.

Properties	Mixture No.	Predicted	Measured	Ratio	Properties	Mixture No.	Predicted	Measured	Ratio
Slump Flow [mm]	1	245.7	253.0	1.02	***f′_tL_*_7_** [MPa]	1	9.0	9.2	1.02
	2	236.5	239.5	1.07		2	9.2	9.5	1.03
	3	217.4	212.5	0.96		3	9.3	8.8	0.95
	4	253.5	257.9	1.03		4	8.6	9.1	1.05
Cone Penetration [mm]	1	33.7	34.5	1.03	***f′_cL_*_7_** [MPa]	1	43.0	40.6	0.95
	2	30.0	32.1	1.06		2	39.0	36.7	0.94
	3	22.3	21.4	0.95		3	37.7	36.2	0.96
	4	35.2	36.4	1.02		4	45.2	42.6	0.94
Yield stress [Pa]	1	1146.1	1187.3	1.04	***f′_tc_*_28_** [MPa]	1	10.0	9.6	0.96
	2	1217.6	1278.9	1.05		2	9.3	9.2	0.99
	3	1327.4	1268.6	0.96		3	9.1	9.4	1.04
	4	894.9	956.3	1.07		4	10.4	9.7	0.93
***f′_tc_*_7_** [MPa]	1	8.6	8.1	0.94	***f′_cc_*_28_** [MPa]	1	59.0	55.8	0.95
	2	8.2	8.4	1.03		2	52.2	54.6	1.05
	3	8.0	8.6	1.07		3	47.7	49.6	1.04
	4	8.8	8.6	0.98		4	60.7	57.4	0.95
***f′_cc_*_7_** [MPa]	1	47.3	51.2	1.08	***f′_tL_*_28_** [MPa]	1	11.7	10.6	0.91
	2	42.2	44.8	1.06		2	11.7	11.3	0.97
	3	41.1	44.7	1.09		3	11.8	11.2	0.95
	4	51.0	55.6	1.09		4	10.9	11.4	1.05
***f′_cL_*_28_** [MPa]	1	50.1	49.8	0.99					
	2	48.5	46.3	0.96					
	3	46.3	43.9	0.95					
	4	52.0	51.7	0.99					

**Table 9 materials-16-01748-t009:** Regression coefficients and *p*-values for all measured properties.

	Intercept	CEM	FA	BA	SP	CEM × FA	CEM × BA	CEM × SP	FA × BA	FA × SP
Slump flow [mm]	226.3	21.9	10.2		29.4	−12.0		−20.9		−10.6
*p*-values		0.0005	0.0495		<0.0001	0.0238		0.0008		0.0418
Penetration [mm]	26.1	5.6	3.9		8.5			−4.8		−3.6
*p*-values		<0.0001	<0.0001		<0.0001			<0.0001		0.0001
Yield stress [Pa]	1122.6		53.8	41.3	−383.9			138.6		185.5
*p*-values			0.0005	0.0038	<0.0001			<0.0001		<0.0001
***f′_tc_*_7_** [MPa]	7.9	0.3			1.1			−0.6		
*p*-values		0.015			<0.0001			<0.0001		
***f′_cc_*_7_** [MPa]	41.4	5.4			5.8			−2.4		
*p*-values		<0.0001			<0.0001			0.0062		
***f′_tL_*_7_** [MPa]	9.2	−0.5		0.5					−0.4	
*p*-values		0.0003		0.0008					0.0018	
***f′_cL_*_7_** [MPa]	39.0	2.8			3.7	2.6				
*p*-values		0.0005			<0.0001	0.0004				
***f′_tc_*_28_** [MPa]	9.4	0.6			1.1			−0.7		
*p*-values		0.0012			<0.0001			0.0009		
***f′_cc_*_28_** [MPa]	50.6	6.1	2.0		5.3					−1.7
*p*-values		<0.0001	0.0488		<0.0001					0.0492
***f′_tL_*_28_** [MPa]	11.5	−0.2	−0.3	0.5		−0.2	0.3			
*p*-values		0.0408	0.0055	<0.0001		0.0092	0.0011			
***f′_cL_*_28_** [MPa]	47.0	4.5	1.5		2.2					−2.5
*p*-values		<0.0001	0.0013		0.0001					<0.0001

**Table 10 materials-16-01748-t010:** Criteria and goals used for multi-objective optimisation.

Criteria	Case 1			Case 2		
	Goal	Weight	Desirability = 0.726	Goal	Weight	Desirability = 0.516
**PARAMETERS**						
CEM (kg/m^3^)	In range	1	650	Minimise	1	614.1
FA (%)	In range	1	11.2	Maximise	2	20
BA (kg/m^3^)	In range	1	3	Maximise	1	3
SP (kg/m^3^)	In range	1	2	In range	1	4
**RESPONSES**						
Slump flow [mm]	Maximise	1	235.2	Maximise	2	247.5
Penetration [mm]	In range	1	35.3	In range	1	33.5
Yield stress [Pa]	Maximise	1	1451.9	Maximise	1	1088.3
***f′_tc_*_7_** [MPa]	In range	1	7.7	In range	1	8.8
***f′_tc_*_28_** [MPa]	In range	1	9.6	In range	1	10.4
***f′_tL_*_7_** [MPa]	In range	1	9.2	Maximise	2	9.1
***f′_tL_*_28_** [MPa]	In range	1	12.2	Maximise	2	11.7
***f′_cc_*_7_** [MPa]	In range	1	43.4	In range	1	48.1
***f′_cc_*_28_** [MPa]	In range	1	50.3	In range	1	57.3
***f′_cL_*_7_** [MPa]	In range	1	37.6	Maximise	1	44.2
***f′_cL_*_28_** [MPa]	In range	1	48.5	Maximise	1	49.4

Note: ‘In range’: range defined in Table 1.

## Data Availability

Not applicable.

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
