# Peer review of "Optimisation of Mix Proportion of 3D Printable Mortar Based on Rheological Properties and Material Strength Using Factorial Design of Experiment"

_materials, 2023, doi:10.3390/ma16041748_

Round 1

Reviewer 1 Report

The article tries to optimize the mix proportion of four selected materials – basalt fibre, fly ash, superplasticizer, and cement - in order to maximize the flowability, yield stress, and compressive and flexural strength of printed mortars. The experimental activity was well-conceived and performed correctly. Nevertheless, some issues are not correctly addressed, and the article needs revisions before being suitable for publication. Some detailed comments are provided below:

·         Abstract: The gap needs to be stated in the abstract of the paper. What is the novelty of the paper?

·         Lines 115-136: The gap in the paper is not clear. The authors should state what is done and what is not in the literature as well as the novelty of the publication.

·         Lines 138-184: I would change the order of sections 2.1 and 2.2. Materials should be first.

·         Line 167: Why did you select those fly ash replacement percentages (5% seems really low)? Also, the cement content is really high compared to common amounts. Why did the authors not use a value lower than 500 kg/m3? Please explain these.

·         Lines 198-220: How did the authors conduct the rheological tests? Was the mortar cast as conventional or did the authors use the nozzle and extruder? If they were cast as usual, I do not know how the authors can relate the results with the mechanical properties. Printing or casting the samples does not give the same flowability, consistency, etc. Please explain this.

·         Lines 325-326: How did the authors fix the issue with the non-printable mortars? Did they include more mixtures? If the authors used statistical modeling to reduce the number of mixtures and they could not make some of them, I am not sure how the authors could extrapolate the other values in between (since the non-printable ones are the ones in the “corners”).

·          Lines 401-696: I feel that there are too many figures in the article. It is hard for the reader to really focus on the main points of the paper. I would move some of the secondary figures and discussion to the supplementary material. Therefore, the main findings of the article will be clearer.

·         Lines 313-696: I am missing more discussion on the paper. The authors just commented on the observations of the experiments instead of making a critical assessment of them. Please include this discussion.

·         Lines 697-747: The conclusions are an overview of the results instead of containing the main findings of the investigation. For instance, conclusions 1 and 2 are not conclusions but observations of the experiments. Based on the expanded discussion, please modify those.

Reviewer 2 Report

The paper is very interesting and deals with the rheological properties and material 10 strength of 3D printable cement mortars. It is well prepared. In my opinion a minor revision is needed:

1. Please add the units in Figures 8, 9 and 10.

2. Please discuss the method of strenght calculations in ccordance to irregular shape of the samples (especially for bending).

3. Please discuss the ststistical distribution of the results - how many samples were tested, are the results repeteable, what about standard deviation etc.

Reviewer 3 Report

The authors have done a good research work, comprehensive and interesting for building professionals. The article deserves to be published if some minor flaws are corrected.

In the abstract and in the paper, care should be taken with the superscripts of m3.

The introduction is well written, contains an appropriate number of references and is correct. Perhaps, the last paragraph could be improved by better highlighting the novelty of this work compared to previous ones.

In section 2, the properties of the materials (e.g. basalt fibres) how they were obtained, if provided by the manufacturer, should be included in the text.

Table 3. In the heading [%] --> Composition [%].

In equations (5), (6) and others, it would be necessary to indicate what each letter in the equation means.

The results are correct and the discussion is appropriate. Perhaps, it would be interesting to study the slip effect of the printed mortar layers under tangential stresses.

Lines 588-595 are out of format.

The limitations of the study should be added in the conclusions.

Sections of the original Template have been deleted and should be included.

The bibliography is not in format.

Reviewer 4 Report

The manuscript titled on "Optimisation of Mix Proportion of 3D Printable Mortar Based 2 on Rheological Properties and Material Strength Using Facto- 3 rial Design of Experiment" is valid. However, the innovative step is not apparent clarified. The major revision is needed.  Remarks are enlisted:

- Basalt fiber needs to add in Title and Keyword.

- Setting time is another key performance of 3DPM. Please address this properties. The fly ash generally reduces the hydration rate leading to slower setting time of mortar. The accelerators like CAC, CSA are introduced. The authors have to discuss in detail on this mix design issue.

- The abbreviations like CEM, w/c, BA. 3DPM were repeated

- Unfortunately, the printing work is done by hand, not printer. Please discuss the research limitation. 

- Figs 2, 3, and 6 should be removed 

- Figure 16: it is necessary to add to develop better image of BA. I think optical microscopy images can better illustrate or using image analysis software.

Round 2

Reviewer 1 Report

The authors have successfully addressed all my comments. I would recommend this article for publication.

Reviewer 4 Report

After read thoroughly the manuscript, I believe the manuscript can be accepted in the current form.